# Pak2 is required for actin cytoskeleton remodeling, TCR signaling, and normal thymocyte development and maturation

Hyewon Phee[1]*, Byron B Au-Yeung[2], Olga Pryshchep[1], Kyle Leonard O'Hagan[1], Stephanie Grace Fairbairn[1], Maria Radu[3], Rachelle Kosoff[3], Marianne Mollenauer[2], Debra Cheng[2], Jonathan Chernoff[3], Arthur Weiss[2,4]

[1]Department of Microbiology-Immunology, Northwestern University Feinberg School of Medicine, Chicago, United States; [2]Department of Medicine, Division of Rheumatology, University of California, San Francisco, San Francisco, United States; [3]Cancer Biology Program, Fox Chase Cancer Center, Philadelphia, United States; [4]Rosalind Russell Medical Research Center for Arthritis, Howard Hughes Medical Institute, University of California, San Francisco, San Francisco, United States

**Abstract** The molecular mechanisms that govern thymocyte development and maturation are incompletely understood. The P21-activated kinase 2 (Pak2) is an effector for the Rho family GTPases Rac and Cdc42 that regulate actin cytoskeletal remodeling, but its role in the immune system remains poorly understood. In this study, we show that T-cell specific deletion of *Pak2* gene in mice resulted in severe T cell lymphopenia accompanied by marked defects in development, maturation, and egress of thymocytes. Pak2 was required for pre-TCR β-selection and positive selection. Surprisingly, Pak2 deficiency in CD4 single positive thymocytes prevented functional maturation and reduced expression of S1P1 and KLF2. Mechanistically, Pak2 is required for actin cytoskeletal remodeling triggered by TCR. Failure to induce proper actin cytoskeletal remodeling impaired PLCγ1 and Erk1/2 signaling in the absence of Pak2, uncovering the critical function of Pak2 as an essential regulator that governs the actin cytoskeleton-dependent signaling to ensure normal thymocyte development and maturation.

*For correspondence: hyewon.phee@northwestern.edu

**Competing interests:** The authors declare that no competing interests exist.

**Reviewing editor**: Fiona M Powrie, Oxford University, United Kingdom

## Introduction

To generate mature T cells that express functional and self-tolerant T cell receptors (TCRs), T cells undergo elaborate developmental selection and maturation processes within the thymus (*Starr et al., 2003*). Signals that result from successful TCR β-chain rearrangement and a pre-TCR formation drive the most immature CD4 and CD8 double negative (DN) thymocytes to become CD4 and CD8 double positive (DP), known as β-selection (*Mallick et al., 1993*). Following TCR α-chain rearrangement and mature TCR expression, DP thymocytes undergo positive selection using their newly assembled TCRs to recognize self peptide-major histocompatibility (pMHC) proteins expressed by the cTECs (cortical thymic epithelial cells). A small number of DP thymocytes successfully undergo positive selection and ultimately mature into CD4 or CD8 single positive (SP) cells (*Scollay and Shortman, 1985*; *Petrie et al., 1990*; *Starr et al., 2003*). TCR-mediated signals that determine thymocyte development are only partially understood.

Following positive selection in the thymic cortex, positively selected thymocytes migrate into the medulla and mature to ensure their functional competency to react to cognate antigens and to survive in the periphery. Fully mature SP thymocytes become competent to proliferate, when stimulated via their TCR, in contrast to DP or semi-mature SP thymocytes that are subject to apoptosis

**eLife digest** T cells are a key element of the immune system. There are many different types of T cells, and they all have their origins in hematopoietic stem cells that are found in the bone marrow. These stem cells leave the bone marrow and circulate in the body until they reach an organ called the thymus, where they become early thymic progenitor cells. These progenitor cells then undergo a process called differentiation to become specific types of T cells, which mature in the thymus before moving to the blood. Although various molecules and mechanisms are known to be involved in the development of T cells, many details of this process are not understood.

One group of molecules that has been implicated in the differentiation of T cells is the p21-activated kinases. Kinases are proteins that activate or deactivate other proteins by adding phosphate groups to specific amino acids. Pak2 adds phosphorylate groups to various proteins that are involved in the reorganization of an important structure inside the cell called the cytoskeleton.

A kinase called Pak2 has an important role in the reorganization of the cytoskeleton, and since this reorganization is involved in almost all aspects of T cell biology, it seems plausible that Pak2 is also involved in the development of T cells. However, it has not been possible to test this idea because deleting the gene for Pak2 in mice results in their death.

Now, Phee et al. have overcome this problem by performing experiments in which the gene for Pak2 was only deleted in T cells. These mice had significantly fewer mature T cells than healthy mice. In particular, the absence of Pak2 in thymocytes (the cells that become T cells) prevented them from maturing into T cells, and also prevented them from producing a receptor protein that is needed for mature T cells to leave the thymus. This work implies that disruption of the Pak2-mediated signaling pathway that regulates the cytoskeleton may weaken the immune system in humans.

(*Hogquist et al., 2005*; *Takada et al., 2011*). Moreover, post-positive selection DP and SP thymocytes increase their IL-7R expression to promote their survival (*Van De Wiele et al., 2004*; *Yu et al., 2006*). As SP thymocytes mature, they decrease expression of CD69 and CD24 and concurrently increase expression of CD62L and Qa2 (*Vernachio et al., 1989*; *Ramsdell et al., 1991*). Therefore, stages of CD4SP thymocyte maturation can be further defined as 'semi-mature' (CD69/CD24$^{hi}$ CD62L/Qa2$^{low}$) and 'mature' (CD69/CD24$^{low}$ CD62L/Qa2$^{hi}$) states. Only functionally mature T cells exit the thymus, which is enabled, in part, by the marked increase in expression of egress receptors, including sphingosine 1 phosphate receptor 1 (S1P1) (*Allende et al., 2004*; *Matloubian et al., 2004*; *Weinreich and Hogquist, 2008*). The transcription factor Kruppel Lung Factor 2 (KLF2) controls expression of S1P1 and CD62L (*Carlson et al., 2006*; *Weinreich and Hogquist, 2008*), thereby governing egress of T cells from the thymus and facilitating entry to the lymph nodes. However, it is not clear which signaling pathways control maturation of SP thymocytes after selection and promote egress by increasing KLF2.

The actin cytoskeleton and proteins that regulate it are involved in almost all aspects of T cell biology. The cytoskeleton, composed of actin filaments, microtubules, and intermediate filaments, provides mechanical support for organization of cellular structure. Reorganization of the cytoskeleton generates dynamic changes in cell shape during cell–cell interaction and migration. TCR engagement drastically induces changes in actin cytoskeletal architecture and reorganizes it. Actin cytoskeletal reorganization influences T cell signaling leading to T cell activation. For instance, TCR-mediated actin polymerization is required to form extensive contacts between T cells and antigen presenting cells (APCs), organize gross cell polarity, and stabilize signaling complexes acting as a scaffold for further assembly (*Wülfing and Davis, 1998*; *Kaizuka et al., 2007*; *Babich et al., 2012*). However, the mechanism that links the actin cytoskeleton to T cell signaling is not well known.

Rho family GTPases such as Rac and Cdc42 regulate cytoskeletal reorganization, microtubule dynamics, and cell polarity in many cell types, including T cells (*Zhao and Manser, 2005*). Activation of Rho family GTPases is tightly regulated by guanine nucleotide exchange factors (GEFs) and GTPase-activating proteins (GAPs) (*Tybulewicz and Henderson, 2009*). Vav family proteins (Vav1, Vav2 and Vav3) are GEFs for Rac and Cdc42 (*Tybulewicz, 2005*). Rac, Cdc42, and Vav play crucial roles in T cell development and activation. For example, mice lack both isoforms of Rac1 and Rac2 or mice deficient in Vav1 show defects in pre-TCR β-selection and positive selection and T cell activation

(*Turner et al., 1997*; *Fujikawa et al., 2003*; *Guo et al., 2008*; *Dumont et al., 2009*; *Tybulewicz and Henderson, 2009*). Cdc42-deficient mice display impaired positive selection and homeostasis of T cells (*Guo et al., 2010*, *2011*). Rac and Cdc42 exert their functions by binding and activating to a large collection of their effector molecules. However, it is not known which protein among many effector molecules of Rac and Cdc42 is required to mediate their defined functions in T cells.

The p21-activated kinases (PAKs), effector molecules of Rac and Cdc42, participate in diverse cellular signaling pathways including cell spreading, adhesion, migration, activation, proliferation, and survival (*Manser et al., 1994*; *Bokoch, 2003*). PAKs are serine/threonine kinases that phosphorylate multiple substrates, including those that are involved in cytoskeletal reorganization, cell proliferation, and survival. The group I PAK family consists of three kinases (Pak1, Pak2 and Pak3) (*Bokoch, 2003*; *Kelly and Chernoff, 2012*). Pak2 is the major PAK expressed in T cells (*Chu et al., 2004*) and has been implicated in T cell function (*Bubeck Wardenburg et al., 1998*; *Yablonski et al., 1998*; *Phee et al., 2005*). However, the in vivo role of Pak2 in T cells is not known due to embryonic lethality of Pak2 knock-out (KO) mice (*Hofmann et al., 2004*; *Kelly and Chernoff, 2012*). Given that Pak2 is one of the well-described effector molecules of Rac and Cdc42, it has long been speculated that Pak2 may mediate the function of Rac and Cdc42 in T cell development and activation. However, considering the fact that Rac and Cdc42 have many effector molecules and may utilize a combination of effectors to mediate a defined cellular function, it was unclear whether Pak2 is the effector molecule that mediates the function of Rac and Cdc42 in T cell development and activation.

In this study, we show for the first time that Pak2 is required for the development, maturation, and timed egress of thymocytes. Lineage-specific deletion of Pak2 at an early T cell developmental stage using the *Lck*-Cre impaired pre-TCR β-selection and positive selection similar to the mice that lack Rac1/2 or Vav1. Surprisingly, deletion of Pak2 at a later developmental stage using the *Cd4*-Cre severely inhibited functional maturation of CD4SP thymocytes, including impaired expression of S1P1 accompanied by reduced expression of its transcription factor, KLF2. TCR-mediated signaling was greatly decreased in the absence of Pak2, including Nur77 induction and S6 phosphorylation, leading to decreased proliferation. Mechanistically, Pak2 deficiency impaired actin cytoskeletal remodeling and activation of PLCγ1 and Erk1/2 triggered by plate-bound TCR stimulation. These findings reveal a critical function of Pak2 as an essential regulator that drives the actin cytoskeleton-dependent signaling for normal thymocyte development and maturation.

## Results

### Generation of T cell-specific Pak2-deficient mice

To examine the role of Pak2 in vivo, we generated Pak2$^{-/-}$ mice using a conventional knock-out (KO) strategy. However, Pak2 deficiency resulted in lethality at embryonic day 8 (*Hofmann et al., 2004*; *Arias-Romero and Chernoff, 2008*; *Kelly and Chernoff, 2012*). To circumvent this problem, we generated conditional Pak2 KO mice by flanking exon 2 of *Pak2* with *loxP* sites (*Kosoff et al., 2013*) and introduced, by breeding, two different Cre transgenes, *Lck*-Cre (*Hennet et al., 1995*) and *Cd4*-Cre (*Lee et al., 2001*), in which the Cre recombinase is expressed under the promoters of T cell-specific *Lck* or *Cd4* genes. The Cre genes in these mice are expressed at different developmental stages. The *Lck*-Cre transgene is expressed during the DN2 (CD4-CD8-double negative 2) stage (*Hennet et al., 1995*), while the *Cd4*-Cre transgene is expressed later at the DP stage (*Lee et al., 2001*). We confirmed that over 95% of Pak2 protein was deleted in DP thymocytes from *Pak2$^{F/F}$;Lck*-Cre or *Pak2$^{F/F}$;Cd4*-Cre mice (*Figure 1A–C*).

### T cell-specific deletion of Pak2 resulted in severe T cell lymphopenia

T-cell specific Pak2 deletion using *Lck*-Cre or *Cd4*-Cre transgenes resulted in severe T cell lymphopenia (*Figure 1D,F,G,J*, *Figure 1—figure supplement 1*). Both CD4 and CD8 peripheral T cells were reduced similarly. Naïve T cells (CD62L$^{hi}$CD44$^{low}$) were almost absent and the few remaining T cells were mostly CD44$^{hi}$ (*Figure 1E,H*). We detected considerable expression of Pak2 in the CD44$^{hi}$ activated (or memory) cells, suggesting that residual CD44$^{hi}$ T cells had undergone lymphopenia-associated expansion from the few T cells that had escaped Cre-mediated deletion (data not shown). Mixed bone marrow chimera (1:1 ratio) experiments revealed that naive T cells generated from *Pak2$^{F/F}$;Cd4*-Cre (*Figure 1I*, *Figure 1—figure supplement 2A–C*) or *Pak2$^{F/F}$;Lck*-Cre (data not shown) hematopoietic stem cells had a cell-intrinsic disadvantage in reconstituting the secondary

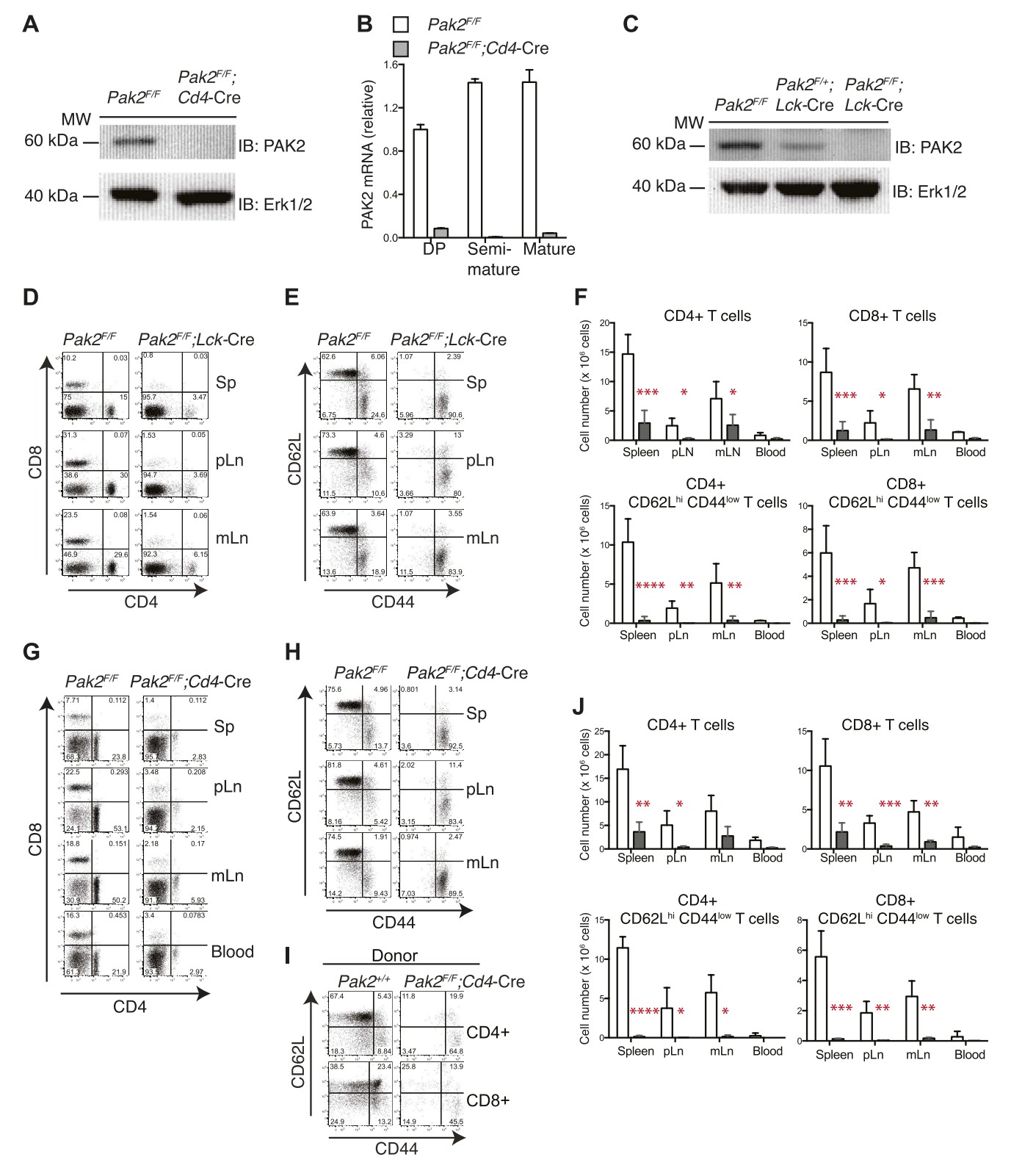

**Figure 1**. T cell lymphopenia in T cell-specific Pak2-deficient mice. (**A**) Western blot analysis using anti-Pak2 antiserum and cell lysates from the thymus of *Pak2^F/F* (WT), and *Pak2^F/F;Cd4*-Cre (KO) mice. Anti-Erk1/2 antibody was used as loading control. Shown are representative of two independent experiments. (**B**) Quantitative PCR analysis of *Pak2* mRNA expression in DP, semi-mature and mature CD4SP thymocytes. Shown are *Pak2* mRNA levels

*Figure 1. Continued*

from DP, semi-mature and mature CD4SP thymocytes relative to *Pak2^F/F* DP thymocytes (error bars; SD). Data are representative of two independent experiments. (**C**) Western blot analysis using anti-PAK2 and cell lysates from the thymi of *Pak2^F/F* (WT), *Pak2^F/+;Lck*-Cre (HET) and *Pak2^F/F;Lck*-Cre (KO) mice. (**D**) Representative flow cytometry analyses of CD4 and CD8 expression on lymphocytes from spleens (n = 5), peripheral lymph nodes (pLn; axillary, brachial and inguinal lymph nodes, n = 4), and mesenteric lymph nodes (mLn, n = 4) from *Pak2^F/F* (WT) and *Pak2^F/F;Lck*-Cre (KO) mice. Numbers in each quadrant represent the percentage of cells in the indicated quadrant. (**E**) Flow cytometry analyses of CD62L and CD44 on CD4+ T cells from *Pak2^F/F* (WT) and *Pak2^F/F;Lck*-Cre (KO) mice. (**F**) Quantification of cell numbers of different lymphocyte subsets from *Pak2^F/F* and *Pak2^F/F;Lck*-Cre mice. Error bars: SD (spleen [n = 5 mice per genotype], pLn [n = 4 mice per genotype], and mLn [n = 4 mice per genotype]; blood [n = 2 mice per genotype]). *, 0.01<p<0.05; **, 0.001<p<0.01; ***, 0.0001<p<0.001; ****, p<0.0001 (unpaired two-tailed Student's t test). (**G**) Representative flow cytometry analyses of CD4 and CD8 expression on lymphocytes from spleen, pLn, mLn, and blood from *Pak2^F/F* (WT) and *Pak2^F/F;Cd4*-Cre (KO) mice. Spleen (n = 4 mice), pLn (n = 4 mice), mLN (n = 3) and blood (n = 2 mice). (**H**) Flow cytometry analyses of CD62L and CD44 within CD4+ T cells from *Pak2^F/F* and *Pak2^F/F;Cd4*-Cre mice. (**I**) Absence of naïve (CD62L^hi CD44^low) CD4 or CD8 T cells generated from *Pak2^F/F;Cd4*-Cre donor bone marrow cells in 1:1 mixed bone marrow chimeras. Data shown are representative of five bone marrow chimeras. (**J**) Quantification of cell numbers of different subsets from *Pak2^F/F* and *Pak2^F/F;Cd4*-Cre mice. Error bars: SD (spleen [n = 4 mice per genotype], pLn [n = 4 mice per genotype], mLn [n = 3 mice per genotype], blood [n = 2 mice per genotype]). *, 0.01<p<0.05; **, 0.001<p<0.01; ***, 0.0001<p<0.001; ****, p<0.0001 (unpaired two-tailed Student's t test). See *Figure 1—figure supplements 1 and 2*.

The following figure supplements are available for figure 1:

**Figure supplement 1**. T cell lymphopenia in T-cell specific Pak2-deficient mice.

**Figure supplement 2**. A T cell intrinsic role of Pak2 in T cell lymphopenia.

lymphoid organs. These results indicate that Pak2 is required for generation or homeostasis of peripheral T cells.

## Deletion of Pak2 at the DN2 stage inhibited β-selection and positive selection

To determine whether T cell lymphopenia in the absence of Pak2 is due to impaired T cell development, we analyzed thymocyte development. Onset of deletion of Pak2 at the DN2 stage in *Pak2^F/F;Lck*-Cre mice inhibited the transition from the DN to DP stage and subsequent positive selection. Total, DP and CD4 and CD8 SP thymocyte numbers were markedly decreased (*Figure 2A,B*). DN3 thymocytes in *Pak2^F/F;Lck*-Cre mice were increased with impaired expression of intracellular TCR β-chain at DN3 but not DN4 stages, suggesting a relative block in development at the checkpoint mediated by the pre-TCR (*Figure 2C,D*). The majority of TCR^hi cells were CD4 or CD8 SP thymocytes in WT mice, but Pak2 deficiency substantially reduced TCR^hi CD4 or CD8 SP thymocytes (*Figure 2E*). To eliminate repertoire selection bias, we evaluated the influence of Pak2 deficiency in the context of the ovalbumin-peptide class II MHC specific OTII TCR transgene (*Barnden et al., 1998*). The generation of OTII TCR transgene+ CD4SP thymocytes was greatly reduced, suggesting defective positive selection (*Figure 2F–H*). In addition, the expression of CD5 was markedly lower in DP and CD4SP thymocytes (*Figure 2I*) with a reduced percentage of post-positive selection cells (data not shown) from OTII+;*Pak2^F/F;Lck*-Cre mice, indicative of lower basal TCR signaling (*Azzam et al., 1998*). Thus, deletion of Pak2 at the DN2 stage markedly impaired T cell development in the thymus by impairing both pre-TCR β-selection and positive selection.

## Deletion of Pak2 at the DP stage inhibited maturation of CD4SP thymocytes

Although we found that Pak2 is required for pre-TCR β selection and positive selection in *Pak2^F/F;Lck*-Cre mice, defects that occurred in DN and DP stages in these mice prevented us from studying the impact of Pak2 on a later developmental stage, namely, the SP stage. To define the effects of Pak2 deficiency at later stages of thymocyte development, we analyzed T cell development in thymi in *Pak2^F/F;Cd4*-Cre mice, in which Pak2 is deleted at the DP stage using the *Cd4* promoter-driven Cre transgene. Numbers of DP and CD4 and CD8 SP thymocytes were similar, suggesting generation of CD4 and CD8 SP thymocytes was normal in *Pak2^F/F;Cd4*-Cre mice (*Figure 3A,B*). Expression of CD5, CD69, and CD3 on DP thymocytes were similar between *Pak2^F/F* and *Pak2^F/F;Cd4*-Cre mice (data not shown). The apparently normal development of DP and SP thymocytes in *Pak2^F/F;Cd4*-Cre mice suggested that Pak2 may be dispensable for positive selection or a defect is masked by a compensatory mechanism in vivo. To eliminate compensatory changes in the TCR repertoire, we introduced the OTII transgene into

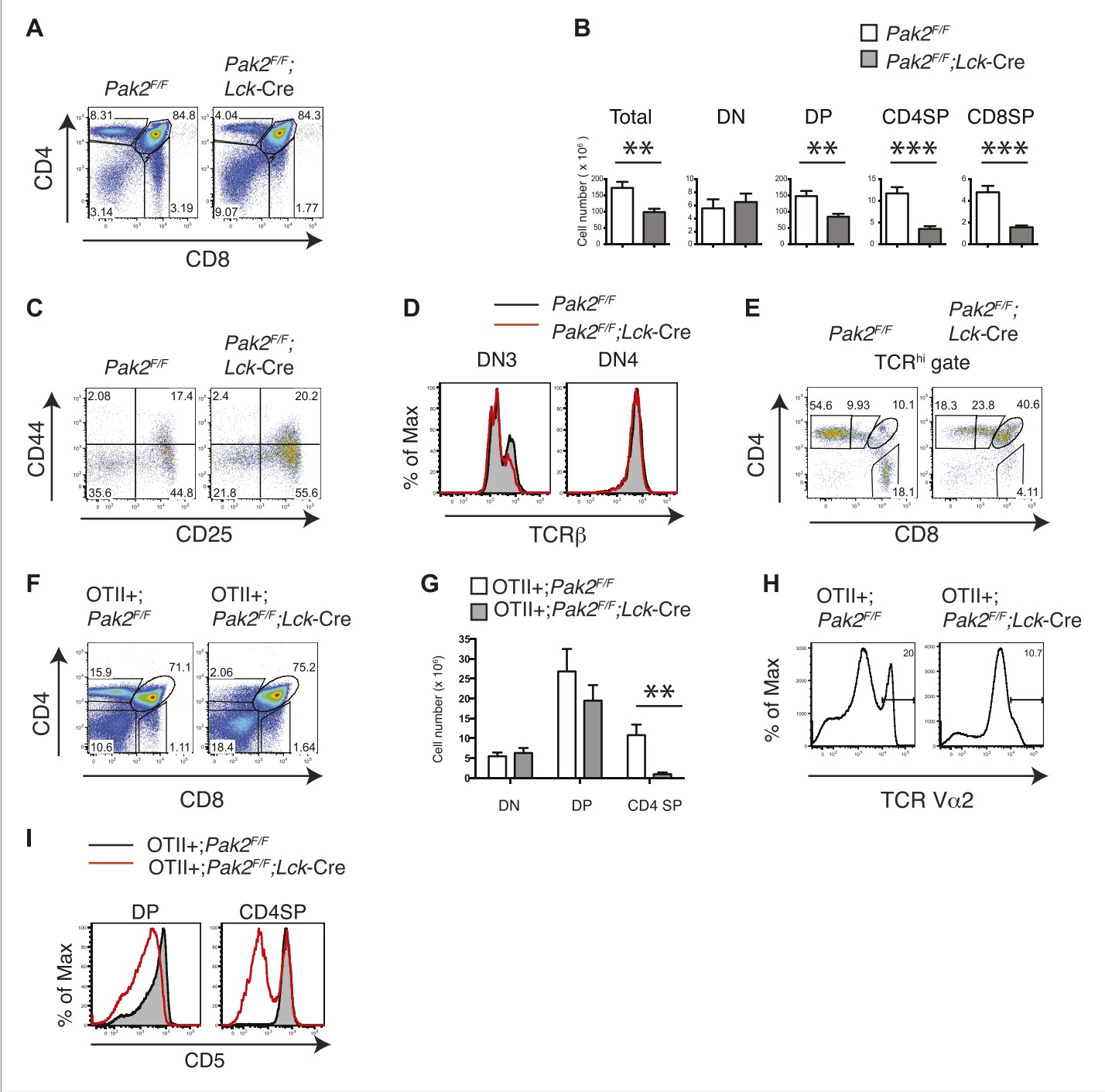

**Figure 2**. Pak2 is required for T cell development. (**A**) Flow cytometry of lymphocytes from thymi of *Pak2^F/F^* or *Pak2^F/F^;Lck*-Cre mice (n = 5). (**B**) Quantification of cell numbers of different thymic subsets from *Pak2^F/F^* and *Pak2^F/F^;Lck*-Cre mice. Error bars: SEM (n = 5 mice per genotype). **, 0.001<p<0.01; ***, 0.0001<p<0.001; (unpaired two-tailed Student's *t* test). (**C**) Expression of CD44 and CD25 on DN thymocytes from *Pak2^F/F^* or *Pak2^F/F^;Lck*-Cre mice. (**D**) Expression of intracellular TCRβ chain in DN3 or DN4 thymocytes. (**E**) Expression of CD4 and CD8 on TCR^hi^ (or CD3^hi^) thymocytes from *Pak2^F/F^* or *Pak2^F/F^;Lck*-Cre mice. (**F**) Percentage of CD4SP thymocytes in OTII^+^;*Pak2^F/F^;Lck*-Cre mice (n = 3). (**G**) Cell numbers of different thymic subsets from OTII^+^;*Pak2^F/F^* or OTII^+^;*Pak2^F/F^;Lck*-Cre mice. Graphs in this figure show mean ± SEM (n = 3). **, 0.001<p<0.01 (**H**) Expression of TCR transgene assessed by anti-Vα2 antibody in total thymocytes from OTII^+^;*Pak2^F/F^* or OTII^+^;*Pak2^F/F^;Lck*-Cre mice. (**I**) Expression of CD5 was reduced on DP thymocytes or CD4SP thymocytes from OTII+;*Pak2^F/F^;Lck*-Cre mice. Shown are representative of three mice per genotype.

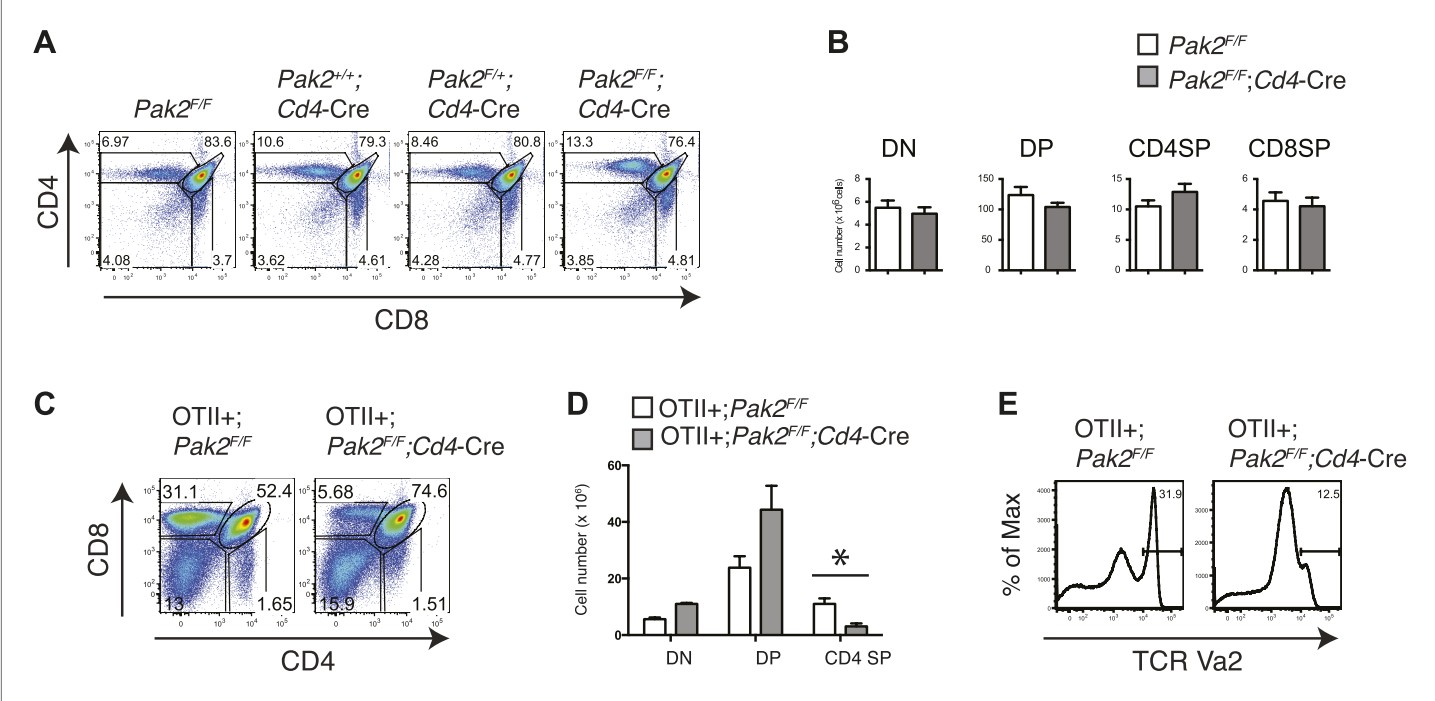

**Figure 3**. Defects in positive selection in OTII+;*Pak2*F/F;*Cd4*-Cre mice. (**A**) Flow cytometry analyses of CD4 SP thymocytes of *Pak2*F/F (WT), *Pak2*+/+;*Cd4*-Cre (WT), *Pak2*F/+;*Cd4*-Cre (Het), or *Pak2*F/F;*Cd4*-Cre (KO) mice. (**B**) Quantification of cell numbers of different thymic subsets. Error bars: SEM (n = 10 mice). (**C**) CD4 and CD8 FACS analyses showing decreased percentage of CD4SP thymocytes in OTII+;*Pak2*F/F;*Cd4*-Cre mice. Shown are representative data from three mice from each genotype. (**D**) Cell numbers of different thymic subsets from OTII+;*Pak2*F/F or OTII+;*Pak2*F/F;*Cd4*-Cre mice. Graphs in this figure show mean ± SEM (n = 3). *, p=0.014. (**E**) Expression of TCR transgene assessed by anti-Vα2 antibody in total thymocytes from OTII+;*Pak2*F/F or OTII+;*Pak2*F/F;*Cd4*-Cre mice. Shown are representative data from three mice from each genotype.

*Pak2*F/F;*Cd4*-Cre mice. The generation of CD4SP thymocytes was markedly reduced in OTII+;*Pak2*F/F;*Cd4*-Cre mice (**Figure 3C–E**), suggesting that Pak2 is required for positive selection.

Severe T cell deficiency in the periphery of *Pak2*F/F;*Cd4*-Cre mice was at odds with the normal numbers of DP and SP thymocytes in *Pak2*F/F;*Cd4*-Cre mice. We hypothesized that Pak2 may play a key role in maturational events after positive selection. CD4SP thymocyte maturation can be further defined by transition from 'semi-mature' (CD69hi CD62Llow) to 'mature' (CD69low CD62Lhi) states (**Gabor et al., 1997**; **Carlson et al., 2006**). Remarkably, CD4SP thymocytes were blocked at the semi-mature stage in *Pak2*F/F;*Cd4*-Cre mice (**Figure 4A,B**). Consistent with a defect at this developmental transition, expression of CD62L and integrin β7 were reduced, but CD69 expression remained high in CD4SP thymocytes (**Figure 4C**, top panels). Expression of CD3, TCRβ, and CD24 were similar between *Pak2*F/F and *Pak2*F/F;*Cd4*-Cre mice (**Figure 4C**, bottom panels), but a subset of *Pak2*F/F;*Cd4*-Cre CD4SP thymocytes exhibited unusually high Qa2 expression (**Figure 4C**, top right panel). However, these Qa2hi CD4SP thymocytes lacked other characteristics of maturation; for example, substantial numbers of thymocytes in this subset maintained high expression of CD69 and CD24 (**Figure 4E**) and they failed to increase CD62L expression (**Figure 4F**).

Since we found that maturation of CD4SP thymocytes was impaired in *Pak2*F/F;*Cd4*-Cre mice, we analyzed the expression of CD69 and CD62L in *Pak2*F/F and *Pak2*F/F;*Lck*-Cre mice. We found that most *Pak2*F/F;*Lck*-Cre CD4SP thymocytes failed to increase CD62L expression and were blocked at the semi-mature stage (**Figure 4—figure supplement 1A**). Moreover, Vα2 TCR transgene positive CD4SP thymocytes from OTII; *Pak2*F/F;*Lck*-Cre mice displayed marked inhibition at the transition from the semi-mature to the mature stage (**Figure 4—figure supplement 1B**). Thus, decreased numbers of CD4SP thymocytes from *Pak2*F/F;*Lck*-Cre or OTII; *Pak2*F/F;*Lck*-Cre mice could be due to decreased positive selection or impaired maturation.

While most DP thymocytes down-regulate expression of IL-7Rα, post-positive selection DP and SP thymocytes increase expression of IL-7Rα (**Van De Wiele et al., 2004**). In the absence of Pak2, IL-7Rα

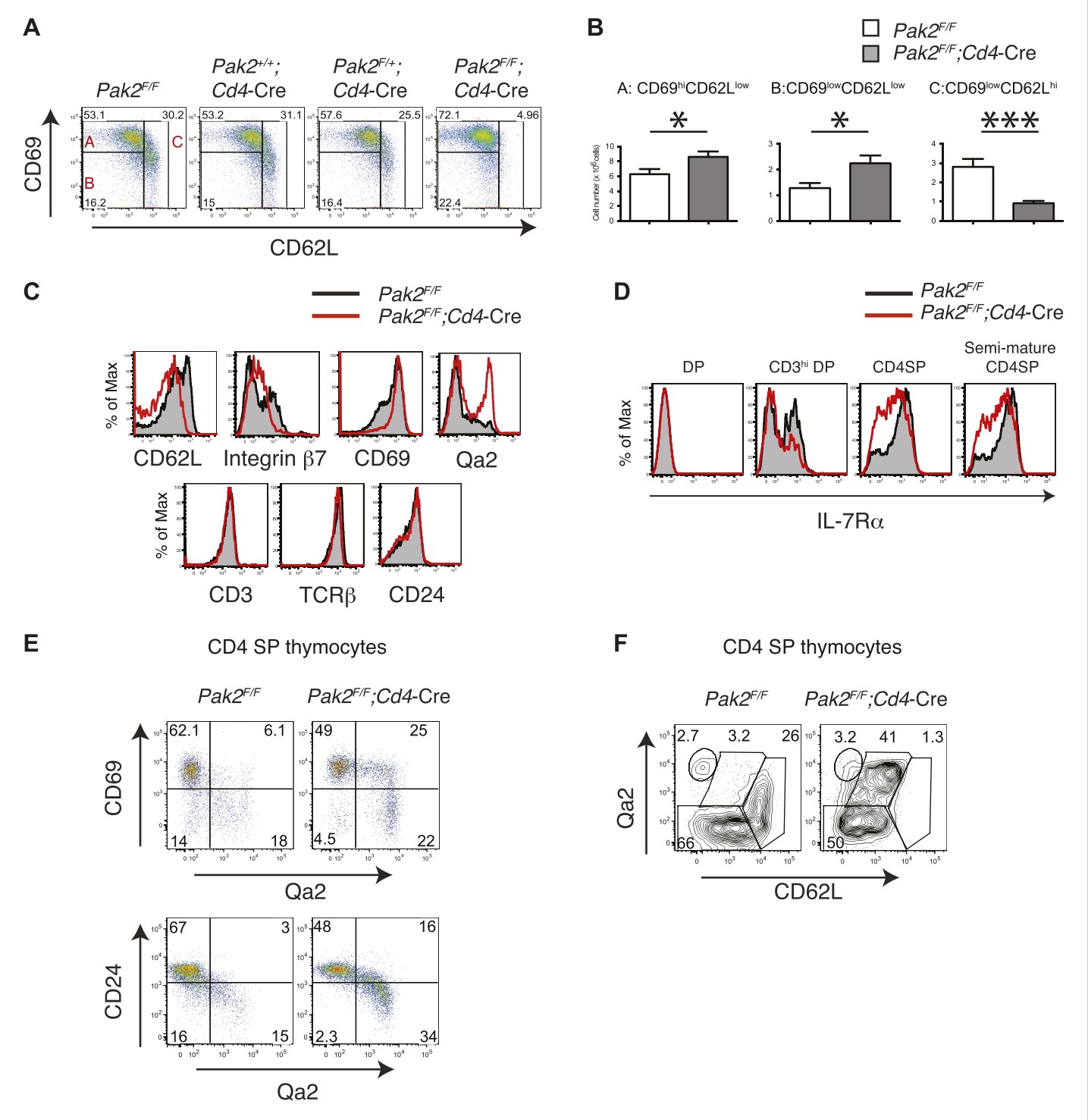

**Figure 4**. Inhibition of the semi-mature to mature transition of CD4SP thymocytes from *Pak2^{F/F}*;*Cd4*-Cre mice. (**A**) CD4SP TCR^{hi} thymocytes were gated and analyzed by expression of CD69 and CD62L. Fraction A (CD69^{hi}CD62L^{low}), semi-mature stage; Fraction B (CD69^{low}CD62L^{low}); Fraction C (CD69^{low}CD62L^{hi}), mature stage. Shown are representative data of ten mice per genotype. (**B**) Quantification of cell numbers of A, B, and C fractions in CD4SP thymocytes. Error bars: SEM (n = 10). *, 0.01<p<0.05; ***, 0.0001<p<0.001 (unpaired two-tailed Student's *t* test). (**C**) Abnormal expression of maturation markers (top panels) in CD4 SP thymocytes. *Pak2^{F/F}* (WT, filled histogram); *Pak2^{F/F}*;*Cd4*-Cre (KO, red). Expression of CD3, TCRβ and CD24 in CD4 SP thymocytes from *Pak2^{F/F}* and *Pak2^{F/F}*;*Cd4*-Cre mice was similar (bottom panels). Shown are representative data of three mice per genotype. (**D**) Decreased expression of IL-7Rα on CD3^{hi}DP, CD4SP and semi-mature (CD69^{hi}CD62L^{low}) CD4SP thymocytes from *Pak2^{F/F}*;*Cd4*-Cre mice. Shown are representative data of three

*Figure 4. Continued on next page*

*Figure 4. Continued*

mice per genotype. (**E**) Abnormal expression of maturation markers in CD4SP thymocytes from *Pak2^F/F^;Cd4*-Cre mice. TCR^hi^ CD4SP thymocytes were gated and analyzed by expression of CD69 vs Qa2 and CD24 vs Qa2. Shown are representative of more than five mice. (**F**) Abnormal expression of maturation markers in CD4SP thymocytes from *Pak2^F/F^;Cd4*-Cre mice. TCR^hi^ CD4SP thymocytes were gated and analyzed by expression of Qa2 vs CD62L. Shown are representative of more than five mice.

The following figure supplements are available for figure 4:

**Figure supplement 1**. Inhibition of the semi-mature to mature transition in CD4SP thymocytes from *Pak2^F/F^;Lck*-Cre or OTII^+^;*Pak2^F/F^;Lck*-Cre mice.

expression was partially decreased in some SP thymocytes, suggesting Pak2 deficiency could affect survival of CD4SP thymocytes (***Figure 4D***). To determine reduced IL-7Rα expression could impact cell survival, we examined whether there was increased cell death or apoptosis by Pak2-deficient cells. The percentages of dead or apoptotic cells detected among resting DP and semi-mature and mature CD4SP thymocytes were comparable from WT and *Pak2^F/F^;Cd4*-Cre mice immediately after harvest (***Figure 5A***) or after 1–2 hr incubation in vitro (***Figure 5B***). However, a twofold increase in cell death was observed in Pak2-deficient semi-mature CD4SP thymocytes when incubated in media for 24 hr, whereas mature CD4SP thymocytes from both WT and Pak2-deficient mice exhibited minimal cell death (***Figure 5C***). Our results suggest that Pak2 plays a key role in the maturation and maintenance of semi-mature CD4SP thymocytes, but not in survival of mature CD4SP thymocytes.

Abnormal expression of maturation markers such as CD62L, integrin β7, Qa2 and IL-7Rα on CD4SP thymocytes from *Pak2^F/F^;Cd4*-Cre mice could be due to T cell-extrinsic factors. For example, the lymphopenic environment of T cell-specific Pak2-deficient mice might alter cytokine profiles in the thymus, which could impact expression of these molecules. To determine the effect of T cell-specific Pak2 deficiency on expression of these maturation markers in a non-lymphopenic environment, we analyzed thymic maturation using competitive bone marrow repopulation experiments. We identified thymocytes generated from WT or *Pak2^F/F^;Cd4*-Cre bone marrow cells using congenic CD45 markers, then compared expression of maturation markers on CD4SP thymocytes (***Figure 6A,B***). We found that expression of CD62L and integrin β7 on CD4SP thymocytes from *Pak2^F/F^;Cd4*-Cre bone marrow donors was greatly decreased. Moreover, down-regulation of CD69 and CD24 was not efficient, suggesting Pak2 deficiency inhibits maturation of CD4SP thymocytes via a thymocyte intrinsic mechanism. Furthermore, CD4SP thymocytes generated from *Pak2^F/F^;Cd4*-Cre bone marrow donors displayed marked reduction in IL-7Rα expression, suggesting that regulation of IL-7Rα by Pak2 is cell autonomous (***Figure 6C***).

## The transition from the semi-mature to the mature stage was blocked in Pak2-deficient CD4SP thymocytes in fetal thymic organ culture

To further determine whether the generation of the mature CD4SP thymocytes was inhibited due to defective maturation, we performed fetal thymic organ culture (FTOC) (***Robinson and Owen, 1977***). Similar to what we observed in *Pak2^F/F^;Cd4*-Cre mice, CD4SP thymocytes from *Pak2^F/F^;Cd4*-Cre FTOC were blocked at the semi-mature stage (***Figure 6D,E***). Interestingly, IL-7Rα expression was similar on DP and CD4SP thymocytes generated from WT and *Pak2^F/F^;Cd4*-Cre FTOCs (***Figure 6F***), suggesting that the observed defects in IL-7Rα expression in adult CD4SP thymocytes from Pak2-deficient mice did not occur in fetal development. Importantly, these results indicate that fetal thymic maturation was still blocked at the semi-mature stage in the absence of Pak2 even when expression of IL-7Rα was not compromised. We conclude that Pak2 is required for transition from the semi-mature to the mature stage in fetal and adult CD4SP thymocytes.

## Pak2 controls expression of KLF2 and S1P1

As SP thymocytes mature and acquire functional competency, they increase the expression of an egress receptor, S1P1 (***Carlson et al., 2006***; ***Weinreich et al., 2009***). The transcription factor KLF2 plays a key role in controlling expression of S1P1 in SP thymocytes (***Carlson et al., 2006***). Expression of KLF2 is controlled during thymocyte development. In DN and DP stages, expression of KLF2 is minimal to prevent premature egress of DN or DP thymocytes. SP thymocytes gradually increase expression of KLF2, reaching highest expression amounts at the mature stage. KLF2 also controls expression of CD62L, which promotes T cell entry to the lymph nodes in the periphery. The signaling pathway that controls timed egress of thymocytes by increasing KLF2 has not been identified. Since

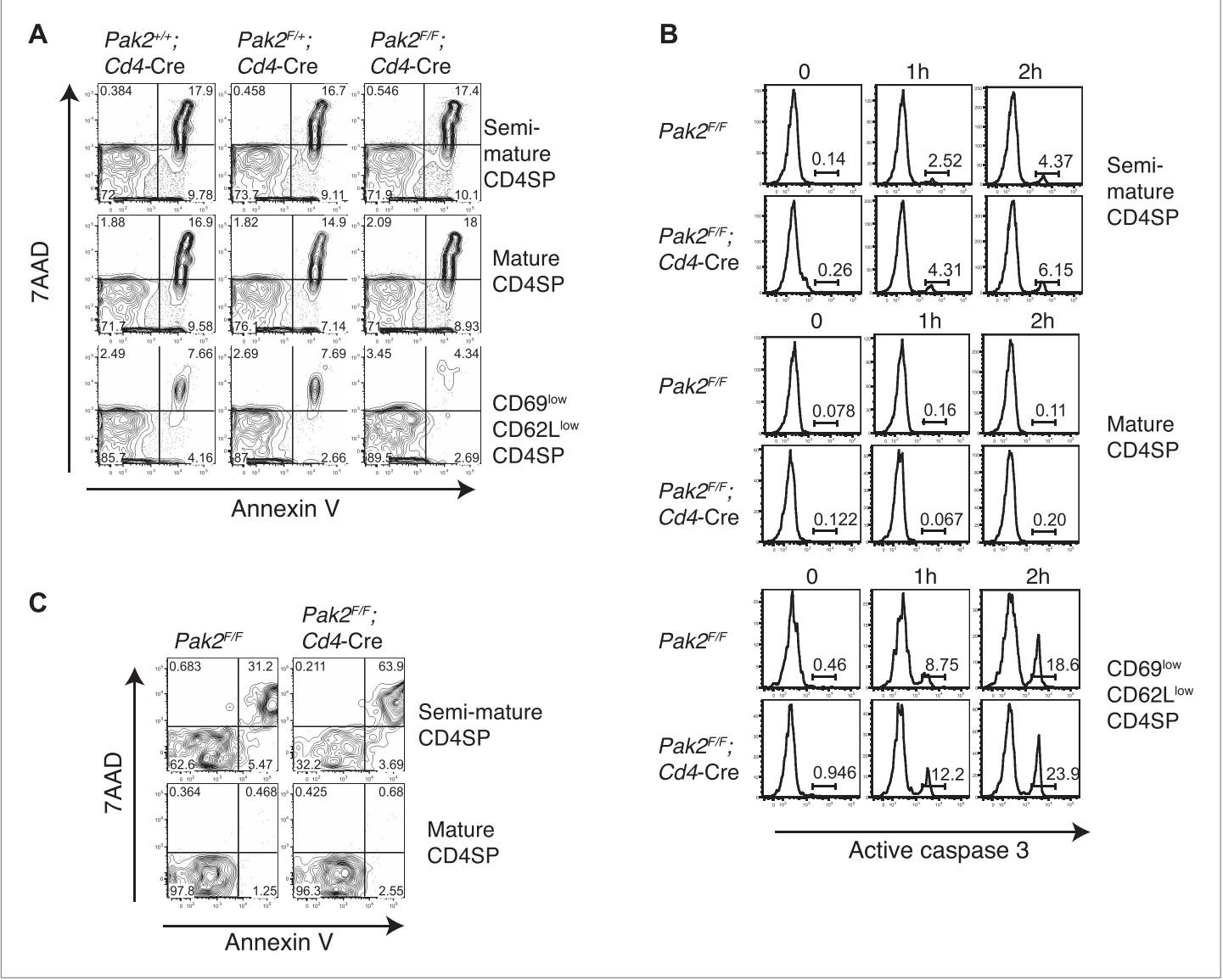

**Figure 5**. Apoptosis or cell death of semi-mature and mature CD4SP thymocytes in freshly isolated cells ex vivo or following in vitro culture. (**A**) Apoptosis detected using 7AAD/Annexin V staining in freshly isolated *Pak2^F/F* or *Pak2^F/F;Cd4*-Cre CD4SP thymocytes. Semi-mature (CD69^hiCD62L^low, Fraction A in *Figure 4A*), mature (CD69^lowCD62L^hi, Fraction C in *Figure 4A*) and CD69^lowCD62L^low (Fraction B in *Figure 4A*) CD4SP thymocytes were gated and analyzed. Data are representative of two independent experiments. (**B**) Intracellular staining of active caspase 3 following 0, 1, and 2 hr of incubation in 10% FBS serum containing media. Data are representative of two independent experiments. (**C**) Apoptosis detected using 7AAD/Annexin V staining following 24 hr of incubation in 10% FBS serum containing media. Semi-mature and mature CD4SP thymocytes were gated and analyzed. Data are representative of two independent experiments.

CD62L expression was reduced in Pak2-deficient CD4SP thymocytes, we asked whether KLF2 expression is altered in the absence of Pak2. Indeed, expression of KLF2 was greatly reduced in CD4SP thymocytes from *Pak2^F/F;Cd4*-Cre mice (**Figure 7A**). Moreover, we found that S1P1 expression was also markedly decreased (**Figure 7B**), suggesting that Pak2-dependent signals regulate timed egress of SP thymocytes by controlling expression of KLF2 and S1P1. Since Pak2 is required for the transition from the semi-mature to the mature stage, the decrease in KLF2 and S1P1 mRNA expression may reflect the lack of the mature CD4SP thymocytes in the absence of Pak2. However, we observed a reproducible decrease in KLF2 expression in the semi-mature CD4SP thymocytes of *Pak2^F/F;Cd4*-Cre mice (**Figure 7A**, right panel), suggesting Pak2 contributes to expression of KLF2 when CD4SP thymocytes are just beginning to upregulate KLF2.

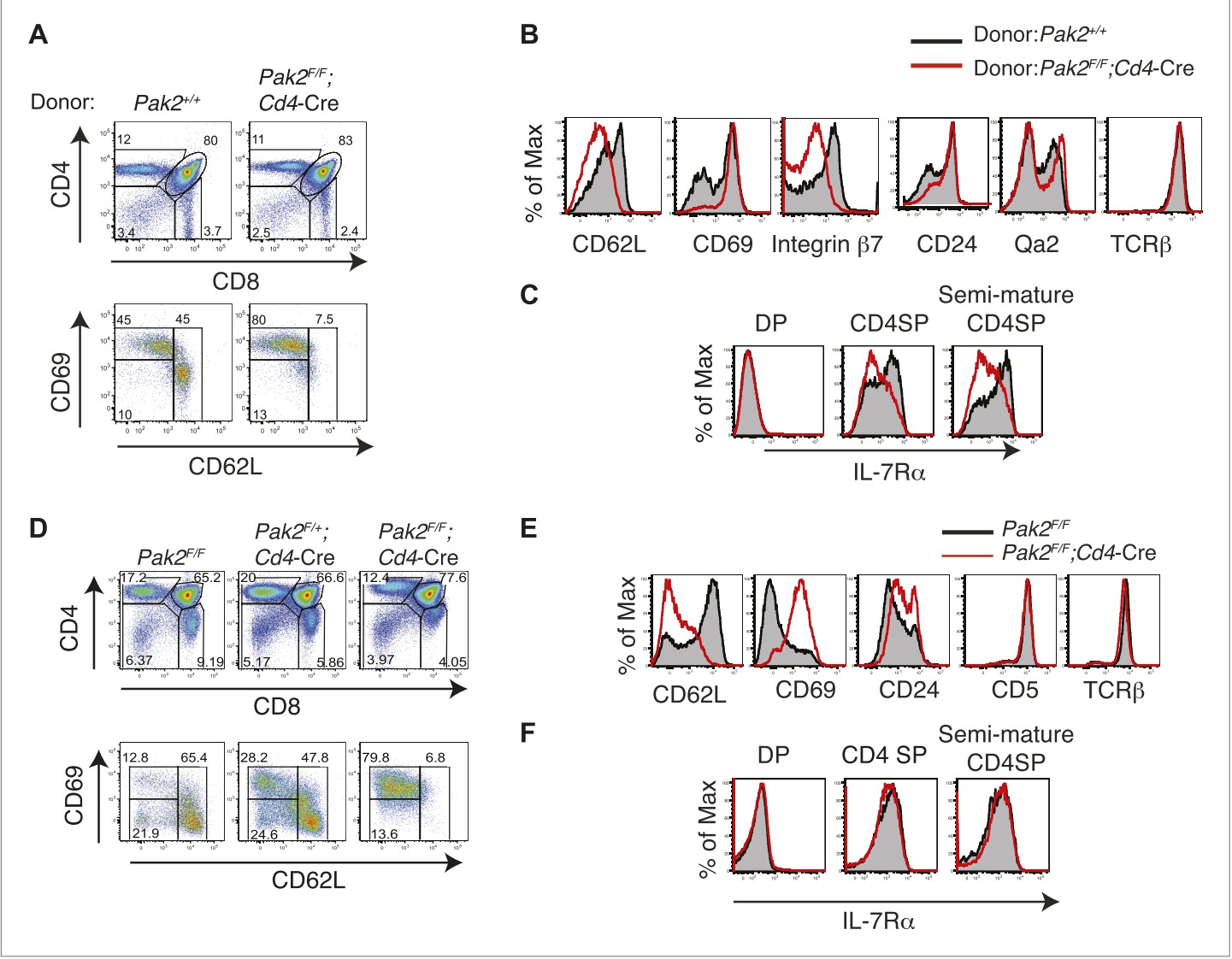

**Figure 6**. Defects in maturation of CD4SP thymocytes in the absence of Pak2 are T cell-intrinsic and Pak2 is required for maturation of fetal CD4SP thymocytes. (**A**) Flow cytometry analysis of 1:1 mixed bone marrow chimeras generated by transferring WT (*Pak2⁺/⁺*, CD45.1⁺CD45.2⁺) and *Pak2^{F/F};Cd4*-Cre mice (CD45.2⁺) donor bone marrow cells that contain hematopoietic stem cells (HSCs) into lethally irradiated C57BL6 hosts that express CD45.1+. Thymocytes generated either from *Pak2⁺/⁺* or *Pak2^{F/F};Cd4*-Cre donor bone marrow cells were identified using CD45 congenic markers and CD4 vs CD8 expression was shown. TCRʰⁱ CD4SP thymocytes were gated and expression of CD69 and CD62L was examined. Data shown are representative of five bone marrow chimeras. (**B**) Aberrant expression of trafficking molecules and maturation markers in CD4 SP thymocytes generated from *Pak2^{F/F};Cd4*-Cre donor. (**C**) Impaired expression of IL-7Rα in CD4SP T cells generated from *Pak2^{F/F};Cd4*-Cre donor bone marrow cells. (**D**) Block in the semi-mature to mature transition of Pak2-deficient CD4 SP thymocytes in fetal thymic organ culture. CD4 vs CD8 (top panels) expression of total culture and CD69 vs CD62L expression within TCRʰⁱ CD4SP population (bottom panels). Shown are data representative FTOC experiments of six embryos per each genotype. (**E**) Altered expression of maturation markers in TCRʰⁱ CD4SP thymocytes from FTOC. (**F**) IL-7Rα expression in DP, CD4SP, and semi-mature (CD69ʰⁱCD62Lˡᵒʷ) CD4SP thymocytes from FTOC.

## Functional maturation of CD4SP thymocytes depends on Pak2

Although CD4SP thymocytes from *Pak2^{F/F};Cd4*-Cre mice lack the CD69ˡᵒʷCD62Lʰⁱ mature subset and did not increase KLF2 or S1P1 expression, it is possible that they are functionally mature. Moreover, within CD4SP thymocytes there is a subset that increases Qa2 expression (*Figure 4E,F*), which could represent a mature subset. Thus, we examined directly whether CD4SP thymocytes from *Pak2^{F/F};Cd4*-Cre mice contain functionally mature cells. One of the hallmarks of functional maturity of CD4SP thymocytes is their ability to proliferate in response to CD3 and CD28 stimulation, unlike semi-mature

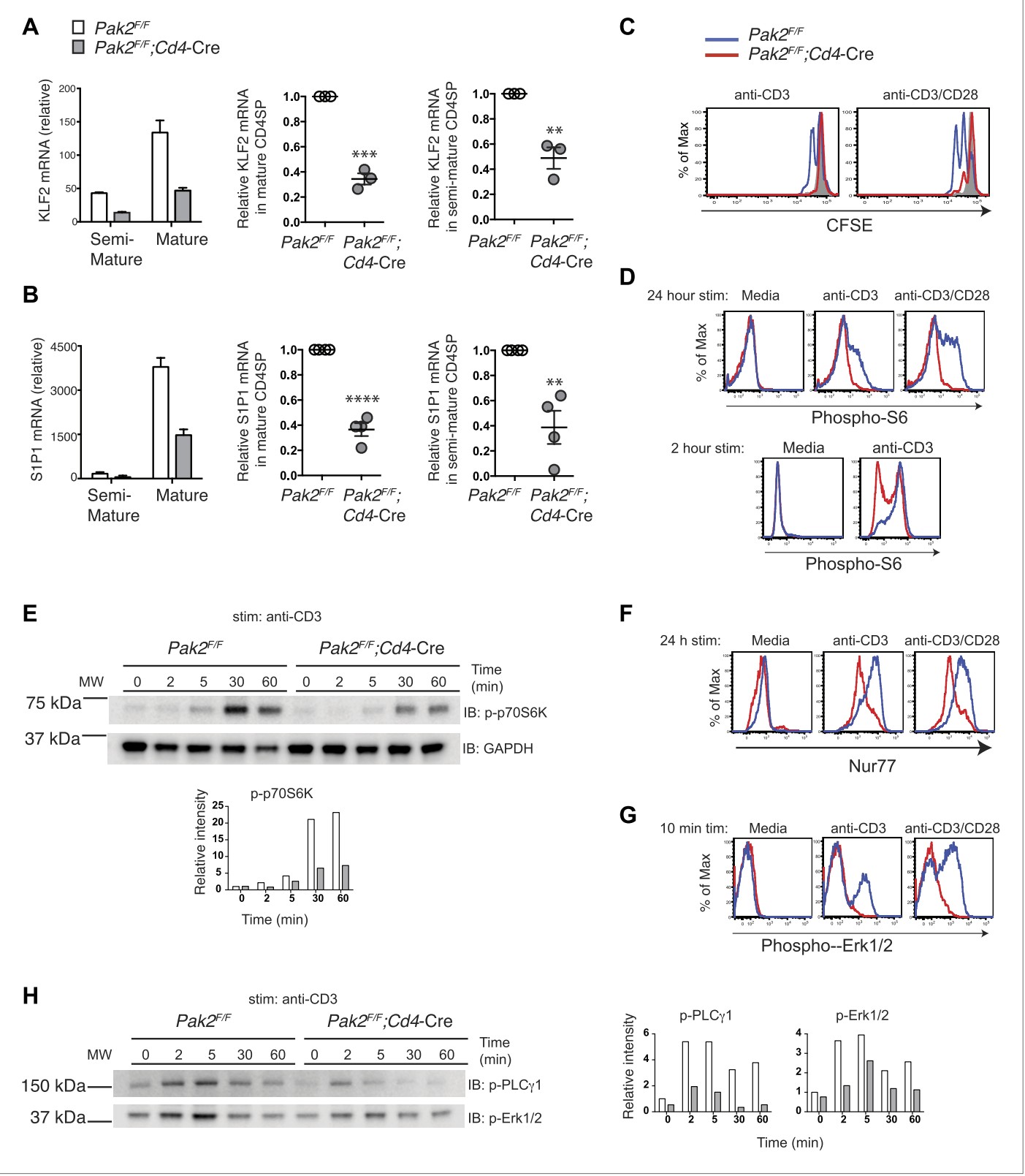

**Figure 7.** Pak2 plays a key role in functional maturation of CD4SP thymocytes. (**A**) Defects in mRNA expression of *KLF2* in mature CD4SP thymocytes from *Pak2^{F/F};Cd4*-Cre mice. *KLF2* mRNA levels in semi-mature and mature CD4SP thymocytes relative to *Pak2^{F/F}* DP thymocytes (left panel, mean ± SD of triplicates, results are representative of three independent experiment); *KLF2* mRNA levels in mature *Pak2^{F/F};Cd4*-Cre CD4SP thymocytes relative to
*Figure 7. Continued on next page*

*Figure 7. Continued*

mature *Pak2^F/F* CD4SP thymocytes (middle panel, mean ± SEM; each dot represents one mouse, n = 3); *KLF2* mRNA levels in semi-mature *Pak2^F/F*;*Cd4*-Cre CD4SP thymocytes relative to semi-mature *Pak2^F/F* CD4SP thymocytes (right panel, mean ± SEM; each dot represents one mouse, n = 3). ***, p=0.0001; **, 0.001<p<0.01. (**B**) Defects in mRNA expression of *S1P1* in mature CD4SP thymocytes from *Pak2^F/F*;*Cd4*-Cre mice. *S1P1* mRNA levels in semi-mature and mature CD4SP thymocytes relative to *Pak2^F/F* DP thymocytes (left panel, mean ± SD of triplicates, results are representative of four independent experiment); *S1P1* mRNA levels in mature *Pak2^F/F*;*Cd4*-Cre CD4SP thymocytes relative to mature *Pak2^F/F* CD4SP thymocytes (middle panel, mean ± SEM; each dot represents one mouse, n = 4); *S1P1* mRNA levels in semi-mature *Pak2^F/F*;*Cd4*-Cre CD4SP thymocytes relative to semi-mature *Pak2^F/F* CD4SP thymocytes (right panel, mean ± SEM; each dot represents one mouse, n = 4). ****, p<0.0001; **, 0.001<p<0.01. (**C**) Proliferative defects of CD4 SP thymocytes from *Pak2^F/F*;*Cd4*-Cre mice. Histograms shown are CFSE dilutions of CD24^low^Qa2^hi CD4SP cells from *Pak2^F/F* and *Pak2^F/F*;*Cd4*-Cre mice following 72 hr of plate-bound anti-CD3 or anti-CD3/CD28 stimulation. *Pak2^F/F*, resting (grey); *Pak2^F/F*, plate-bound anti-CD3 or anti-CD3/CD28 stimulation (blue histogram); *Pak2^F/F*;*Cd4*-Cre, plate-bound anti-CD3 or anti-CD3/ CD28 stimulation (red histogram). Shown are data representative of two independent experiments. (**D**) Phosphorylation of S6 in media or following plate-bound antibody stimulation for 24 hr (top panels) and 2 hr (bottom panels) in total CD4SP thymocytes. Shown are data representative of three (24 hr) or two (2 hr) independent experiments. *Pak2^F/F*, blue; *Pak2^F/F*;*Cd4*-Cre, red histogram. (**E**) Immunoblotting analysis of phosphorylation status of p70S6K (T389) in total thymocytes from *Pak2^F/F* or *Pak2^F/F*;*Cd4*-Cre mice in media or following plate-bound anti-CD3 stimulation for 2, 5, 30 and 60 min. Intensity of each band was measured, normalized by intensity of GAPDH (as loading control), and shown in the graph as relative intensity. Shown are data representative of three independent experiments. (**F**) Induction of Nur77 following plate-bound antibody stimulation for 24 hr. Shown are data representative of three independent experiments. *Pak2^F/F*, blue; *Pak2^F/F*;*Cd4*-Cre, red histogram. (**G**) Phosphorylation of Erk1/2 in CD4SP cells in media or following plate-bound antibody stimulation for 10 min. Shown are data representative of three independent experiments. *Pak2^F/F*, blue; *Pak2^F/F*;*Cd4*-Cre, red histogram. (**H**) Immunoblotting analysis of phosphorylation status of PLCγ1 (Y783) and Erk1/2(T202/Y204) in total thymocytes from *Pak2^F/F* or *Pak2^F/F*;*Cd4*-Cre mice in media or following plate-bound anti-CD3 stimulation for 2, 5, 30 and 60 min. Intensity of each band was measured, normalized by intensity of GAPDH (shown in *Figure 7E*), and shown in the graph as relative intensity. Shown are data representative of three independent experiments. See *Figure 7—figure supplements 1 and 2*.

The following figure supplements are available for figure 7:

**Figure supplement 1**. Increased apoptosis or cell death in semi-mature CD4SP thymocytes from *Pak2^F/F*;*Cd4*-Cre mice following 24 hr in vitro culture.

**Figure supplement 2**. mTOR-dependent phosphorylation of p70S6K and S6.

thymocytes that are more susceptible to cell death (*Hogquist et al., 2005*; *Takada et al., 2011*). We compared the proliferative responses of CD24^low^Qa2^hi CD4SP cells from WT and *Pak2^F/F*;*Cd4*-Cre mice following plate-bound TCR and CD28 stimulation. As expected, the CD24^low^Qa2^hi CD4SP thymocytes from WT mice underwent 2–3 rounds of cell division following CD3 or CD3 and CD28 stimulation, suggesting these cells are functionally mature. In contrast, most CD4SP thymocytes from *Pak2^F/F*;*Cd4*-Cre mice including the CD24^low^Qa2^hi subset did not proliferate following TCR/CD28 stimulation (*Figure 7C*). In addition, the Qa2^hi CD4SP thymocytes from *Pak2^F/F*;*Cd4*-Cre mice were more susceptible to cell death following stimulation (*Figure 7—figure supplement 1A*), suggesting that these cells are not functionally mature. Of note, Pak2-deficient CD4SP thymocytes are more susceptible to cell death following 24 hr of incubation in resting or stimulated conditions because they contain more semi-mature thymocytes that undergo cell death (*Figure 7—figure supplement 1B–D*). Together, these results indicate that functional maturity of CD4SP thymocytes depends on Pak2.

Since we found that Pak2 is required for proliferative response of CD4SP thymocytes following plate-bound anti-CD3 or anti-CD3/CD28 stimulation, we sought to determine the mechanism by which Pak2 affects signaling pathways activated by the TCR and CD28 stimulation. The PI3K/Akt/mTORC1 signaling pathway has been reported to be activated by the TCR and CD28 stimulation, reflected in the phosphorylation of S6 ribosomal protein (S6), a direct target of the mTORC1-activated p70S6 kinase (p70S6K) (*Hay and Sonenberg, 2004*). Activation of the mTORC1 pathway increases rates of mRNA translation and protein synthesis, which permits increased cell growth and proliferation (*Mills and Jameson, 2009*). Activation of mTORC1 pathway was absolutely required for phosphorylation of S6 at the S235/S236 sites and p70S6K at the T389 site, as phosphorylation of these sites were completely abrogated by rapamycin, an inhibitor of mTOR (*Figure 7—figure supplement 2B–E*). Phosphorylation of S6 at S235/S236 in CD4SP thymocytes from WT mice was increased after 24 hr of stimulation (*Figure 7D*, top panels). On the contrary, most of the CD4SP thymocytes from *Pak2^F/F*;*Cd4*-Cre mice displayed minimal S6 phosphorylation. Since TCR-induced S6 phosphorylation reaches its optimal amounts after 1 or 2 hr of TCR stimulation and lasts up to 5 hr (*Salmond et al., 2009*; *van den Brink et al., 1999*), we examined S6 phosphorylation following 2 hr of plate-bound anti-CD3 stimulation. Phosphorylation of S6 at S235/S236 was substantially decreased in the absence of Pak2 (*Figure 7D*,

bottom panels). Activation of MAPK-RSK pathway contributes to phosphorylation of S6 (*Salmond et al., 2009*). Indeed, treatment of thymocytes with a MEK inhibitor, UO126, partially reduced phosphorylation of S6 at S235/S236 (*Figure 7—figure supplement 2C–E*). However, since rapamycin treatment completely abrogated phosphorylation of S6 at these sites (*Figure 7—figure supplement 2C–E*), the MAPK-dependent phosphorylation of S6 also seems to depend on activation of mTORC1-mediated pathway. To determine whether reduced phosphorylation of S6 in the absence of Pak2 is due to inhibition of mTORC1-mediated pathway, we examined phosphorylation of p70S6K at the T389 site because phosphorylation of this site was completely dependent on mTORC1-mediated pathway but independent of MAPK-mediated pathway (*Figure 7—figure supplement 2C–E*). Phosphorylation of p70S6K at T389 was also decreased following 30 min and 1 hr of plate-bound CD3 stimulation in the absence of Pak2 (*Figure 7E*). These findings demonstrate that Pak2 participates in mTORC1-mediated activation of p70S6K and phosphorylation of S6 following plate-bound TCR stimulation.

To determine whether Pak2-deficient CD4SP thymocytes have a defect in optimal activation of integrated TCR-mediated signaling events, we examined expression of Nur77, an orphan nuclear hormone receptor induced in response to TCR stimulation, as a reporter for TCR strength of signaling (*Moran et al., 2011*; *Zikherman et al., 2012*). Nur77 induction was greatly decreased in CD4SP thymocytes from *Pak2^{F/F};Cd4*-Cre mice following 24 hr of anti-CD3 or anti-CD3/CD28 stimulation (*Figure 7F*). These results reveal that Pak2 is required for downstream events reflective of TCR signaling, including those inducing Nur77 induction and p70S6K activation.

## Pak2 is required for optimal activation of PLCγ1 following plate-bound TCR stimulation

Decreased induction of Nur77 following TCR or TCR/CD28 stimulation suggests that Pak2 is required for TCR-dependent signaling events. Since Pak2 is required for multiple processes of development and maturation of thymocytes, it is possible that Pak2 regulates these processes by governing TCR-mediated signaling. Because activation of Erk1/2 following TCR stimulation plays a key role in thymic development as well as in induction of Nur77 (*van den Brink et al., 1999*), we examined activation of Erk1/2 following TCR or TCR/CD28 stimulation. Phosphorylation of Erk1/2 was markedly reduced in the absence of Pak2, when cells were stimulated by plate-bound anti-CD3 or anti-CD3/CD28 stimulation (*Figure 7G,H*). In contrast, phosphorylation of Erk1/2 was not reduced when cells were stimulated by soluble anti-CD3 or anti-CD3/CD28 in the absence of Pak2 (data not shown), suggesting that Pak2 is required for transducing signaling events that are specifically provoked by plate-bound TCR stimulation. TCR engagement by plating T cells on surfaces coated with antibodies to the TCR mimics T cell activation by antigen-presenting cells (APCs) and triggers dynamic changes in actin cytoskeletal architecture, such as formation and expansion of contacts bounded by continuously remodeled actin-rich rings (*Wear et al., 2000*; *Bunnell et al., 2001*; *Babich et al., 2012*). These actin-rich rings are associated with extension of lamellipodia that requires assembly and disassembly of actin filament neworks (*Bunnell et al., 2001*). Changes in actin cytoskeletal architecture induced by TCR engagement promote T cell signaling (*Babich et al., 2012*). Interestingly, actin dynamics that mediate ongoing actin polymerization and retrograde flow promotes sustained PLCγ1 signaling during plate-bound TCR activation, while disruption of F-actin dynamics inhibits sustained PLCγ1 phosphorylation (*Babich et al., 2012*). Since activation of PLCγ1 plays a critical role in activating Erk1/2 following TCR stimulation, we hypothesized that Pak2 participates in activation of Erk1/2 via promoting PLCγ1 activation by regulating actin cytoskeletal dynamics during T cell spreading triggered by TCR engagement. Indeed, we found that phosphorylation of PLCγ1 that is known to be associated with its activation status (*Kim et al., 1991*) is profoundly decreased in the absence of Pak2, indicating that Pak2 is required for optimal PLCγ1 activation, possibly by regulating actin cytoskeletal reorganization (*Figure 7H*).

## Pak2 is required for actin cytoskeletal remodeling during T cell spreading triggered by plate-bound TCR stimulation

Our results suggest that Pak2 may participate in actin cytoskeletal dynamics during T cell spreading triggered by TCR stimulation. To determine whether the loss of Pak2 affects the actin cytoskeleton in CD4SP thymocytes, we first examined actin polymerization and Rac1 activation in resting conditions. To our surprise, the amount of polymerized filamentous actin (F-actin) was greatly increased in resting conditions (*Figure 8A*), suggesting processes regulating actin assembly and disassembly were disrupted in the absence of Pak2. Consistent with these data, Rac1 activation was consistently increased

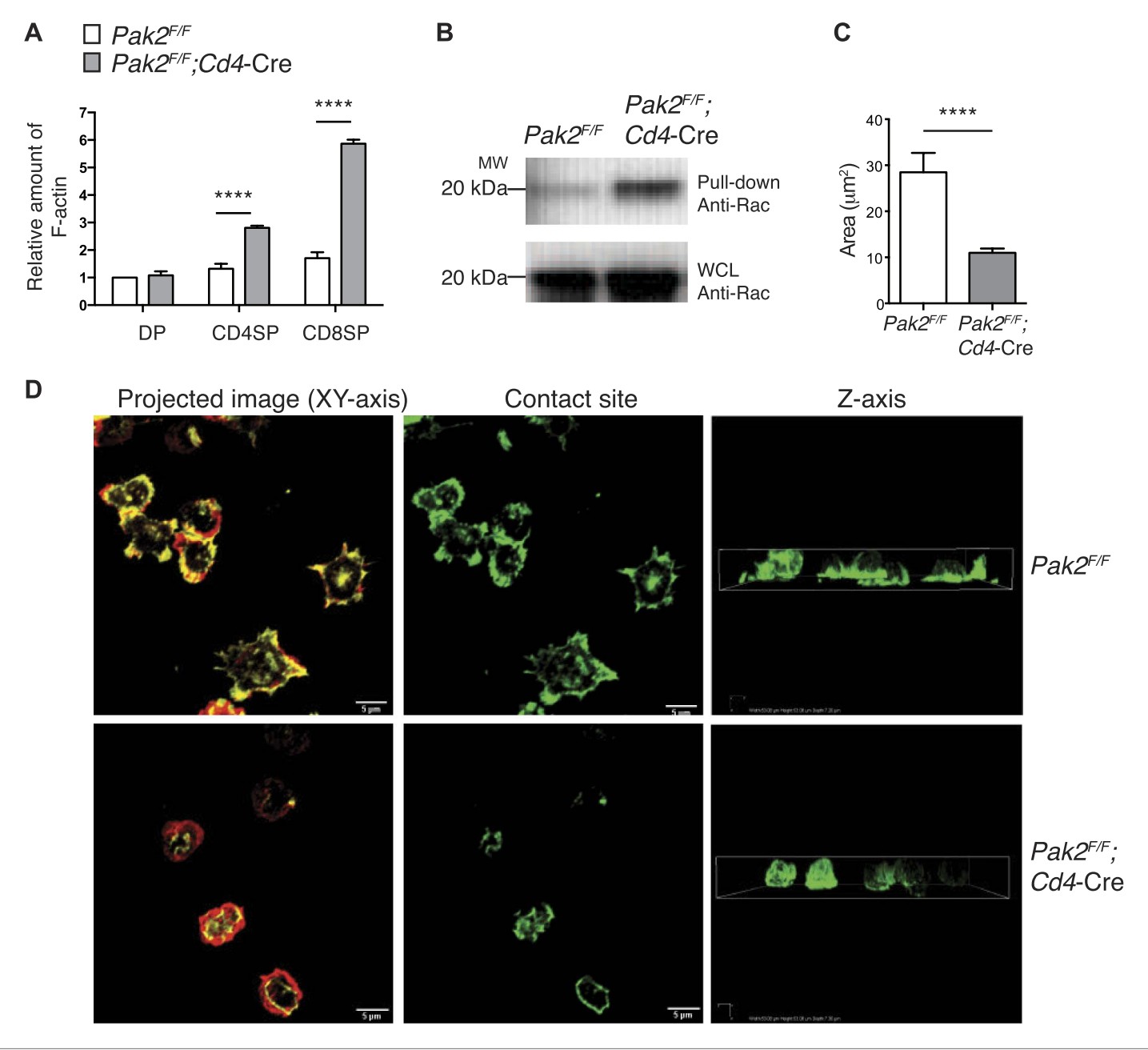

**Figure 8**. Pak2 regulates actin dynamics of CD4SP thymocytes. (**A**) Increased actin polymerization in resting DP, CD4 and CD8 SP thymocytes from *Pak2*[F/F]*Cd4*-Cre mice. The mean fluorescence intensity of phalloidin-Alexa488 in the DP, CD4SP and CD8SP thymocyte populations was normalized to the value of WT DP thymocytes at rest. (error bars; SEM, n = 3). ****, p<0.0001 (**B**) Increased activation of Rac1 at resting from *Pak2*[F/F]*;Cd4*-Cre thymocytes using Rac1 pull-down assay. WCL; whole cell lysates. Shown are data representative of three independent experiments. (**C**) Spreading areas of CD4SP thymocytes on the cover slips. Cells were allowed to spread for 60 min on coverslips coated with anti-CD3. Z-stack images of F-actin staining of the cells were collected and spreading areas where cells contact the cover slips were analyzed. Shown are data representative of three independent experiments. (error bars; SEM, *Pak2*[F/F] [n = 21], *Pak2*[F/F]*;Cd4*-Cre [n = 25]). ****, p<0.0001. (**D**) T cell spreading and actin polymerization triggered by plate-bound TCR stimulation. Shown are z-stack images of the cells from the contact sites where T cells touch the cover slides to the top of the cells. Left panels; maximal projection of z-stack images of F-actin staining (F-actin staining at the contact site [green] and F-actin staining of the rest of the z-stacks [red]). Scale bar: 5 μm. Middle panels: F-actin staining on the contact site to visualize cell spreading. Right panels: z stack images of F-actin were complied and reconstructed as a 3D image. Shown is the orthogonal image viewed from the z-axis. Shown are data representative of three independent experiments.

under resting conditions (*Figure 8B*). However, despite increased F-actin and activated Rac levels, Pak2-deficient CD4SP thymocytes did not reorganize their actin cytoskeleton and spread when triggered by plate-bound TCR stimulation. WT CD4SP thymocytes exhibited intense spreading at 60 min of stimulation, when they were plated on the coverslips coated with antibodies to the TCR (*Figure 8D*, upper left and middle panels), while polarizing F-actin of the cell body towards to the contact sites (*Figure 8D*, upper right panel). In stark contrast, Pak2-deficient CD4SP thymocytes displayed poor spreading at the contact sites and actin polymerization at the contact sites was markedly impaired (*Figure 8D*, lower left and middle panels), while F-actin in the cell was not polarized towards to the contact sites (*Figure 8D*, lower right panel). As a result, areas where cells spread at the contact sites were significantly reduced in the absence of Pak2 (*Figure 8C*). These findings revealed that Pak2 plays a vital role in regulating the actin cytoskeletal reorganization induced by TCR stimulation. Since the integrity of dynamic actin cytoskeletal structure and actin cytoskeletal reorganization triggered by TCR engagement are fundamental for the proper signal transmission in many biological processes of T cells, we propose that Pak2 controls multiple stages of T cell development in the thymus by governing dynamic rearrangement of the actin cytoskeleton.

## Discussion

We report that Pak2, an effector for the small GTPase Rac and Cdc42, is essential for multiple stages of thymocyte development and maturation. T cell-specific deletion of Pak2 at two different developmental stages using stage-specific Cre transgenic mice demonstrated that Pak2 is critical for T cell development, maturation and egress. We found that deletion of Pak2 at either the DN2 or DP stages using the *Lck*-Cre or *Cd4*-Cre transgenic mice resulted in severe T cell lymphopenia. Development of DP and SP thymocytes was inhibited in *Pak2^F/F^;Lck*-Cre mice due to impaired pre-TCR β-selection and positive selection, resulting in reduced numbers of DP and SP thymocytes. In contrast, generation of SP thymocytes was not affected, nor were numbers of DP and SP thymocytes reduced in *Pak2^F/F^;Cd4*-Cre mice. Apparent normal development was due to compensatory changes in the TCR repertoire because positive selection was impaired when the OTII transgene was introduced in *Pak2^F/F^;Cd4*-Cre mice. Pak2 deletion in *Pak2^F/F^;Cd4*-Cre mice presented a unique opportunity to decipher the role of Pak2 in maturation of CD4SP thymocytes after positive selection since generation of CD4SP thymocytes was not impaired. We found that Pak2 is required for the functional maturation and timed egress of CD4SP thymocytes in *Pak2^F/F^;Cd4*-Cre mice.

To define the mechanisms by which Pak2 regulates thymocyte development and maturation, we sought to determine Pak2's function in TCR-induced signaling events. We found that Pak2 is required to remodel actin cytoskeletal architecture during T cell spreading induced by TCR stimulation. Since TCR engagement induces rapid changes in cytoskeletal reorganization in T cells, which is critical for transducing signals, it is possible that perturbation of the actin cytoskeletal reorganization processes in Pak2-deficient thymocytes prevents normal TCR-mediated signaling events. Indeed, inability to remodel the actin cytoskeleton during T cell spreading was accompanied by impaired activation of PLCγ1 and Erk1/2, suggesting Pak2 is required to drive actin cytoskeleton-dependent signaling induced by the TCR. As a result, induction of Nur77 and phosphorylation S6 were inhibited, and proliferative responses were diminished in the absence of Pak2. Activation of PLCγ1 (*Sommers et al., 2005*; *Fu et al., 2010*) and Ras-mediated MAPK pathway leading to Erk1/2 activation are essential for positive selection (*Alberola-Ila et al., 1995*; *Swan et al., 1995*; *O'Shea et al., 1996*; *Swat et al., 1996*; *Pagès et al., 1999*; *Fischer et al., 2005*; *McGargill et al., 2009*). Since positive selection is initiated by TCR/pMHC interaction between DP thymocytes and cTECs and controlled by avidity and affinity of this interaction (*Starr et al., 2003*), inefficient signaling such as impaired activation of PLCγ1 and Erk1/2 could lead to defects in positive selection in the absence of Pak2. We also confirmed that disruption of actin dynamics by administration of either the F-actin stabilizing agent jasplakinolide (*Bubb et al., 1994*; *Murthy and Wadsworth, 2005*) or the actin depolymerizing agent latrunculin (*Spector et al., 1983*; *Coué et al., 1987*) completely abrogated signaling events such as induction of Nur77 and S6 phosphorylation following plate-bound TCR stimulation (data not shown), suggesting actin cytoskeletal reorganization is critical for activation of TCR-induced signaling events. Together, these findings suggest that Pak2 is required to coordinate T cell signaling network by controlling actin cytoskeletal reorganization during thymocyte development.

After completing positive selection, post-positive selection DP thymocytes migrate from the thymic cortex to the medulla and spend about 4–5 days there prior to exiting (*McCaughtry et al., 2007*).

Newly selected SP thymocytes acquire 'mature' features in the medulla before they egress (*McCaughtry et al., 2007*). During this time, they decrease expression of CD69 and CD24 and increase expression of CD62L, integrin β7, and Qa2. In addition, SP thymocytes become functionally mature as they acquire competency to proliferate in response to TCR stimulation, rather than die like their immature precursors DP or semi-mature SP thymocytes. We found that CD4SP thymocytes from *Pak2^{F/F};Cd4*-Cre mice were arrested at the semi-mature stage, unable to increase expression of CD62L and integrin β7. CD4SP thymocytes from *Pak2^{F/F};Cd4*-Cre mice did not contain a functionally mature subset because they failed to proliferate following CD3 or CD3/CD28 stimulation. Rather, CD4SP thymocytes from *Pak2^{F/F};Cd4*-Cre mice contain more semi-mature cells that undergo cell death when stimulated, confirming that Pak2 is required for functional maturation of CD4SP thymocytes. The effect of Pak2 deficiency in CD4-expressing SP thymocytes was specific to the transition from the semi-mature to the mature stage after positive selection, since the generation of semi-mature CD4SP thymocytes from DP thymocytes was normal in *Pak2^{F/F};Cd4*-Cre mice. The mechanism by which Pak2 mediates the transition from the semi-mature to the mature stage of CD4SP thymocytes is not clear. Since semi-mature CD4SP thymocytes from *Pak2^{F/F};Cd4*-Cre mice were more prone to cell death following 24 hr of in vitro culture, Pak2 may be required for maintenance of semi-mature CD4SP thymocytes. However, Pak2 deficiency in CD4-expressing SP thymocytes did not increase cell death or apoptosis in freshly isolated thymocytes ex vivo or after 1–2 hr of in vitro culture. Moreover, the number of semi-mature CD4SP thymocytes was not reduced in *Pak2^{F/F};Cd4*-Cre mice, suggesting that the effect of Pak2 deficiency may be specific to the semi-mature CD4SP thymocytes that received a maturation signal. For example, the maturation signal may activate Pak2 in semi-mature cells to provide protection from apoptosis and to make the transition to the mature stage. The nature of the upstream pathway that controls maturation and is also dependent upon Pak2 is currently unknown.

Another key feature linked to SP thymocyte functional maturation is the increased expression of the egress receptor S1P1 as a result of expression of KLF2, a crucial transcription factor that regulates expression of S1P1 and CD62L (*Carlson et al., 2006*; *Weinreich and Hogquist, 2008*). Failure to generate mature CD4SP thymocytes in the absence of Pak2 greatly impeded both KLF2 and S1P1 expression. Although KLF2 plays a critical role in egress and trafficking, KLF2 does not contribute to functional maturation of thymocytes because KLF2-deficient SP thymocytes are not defective in proliferation following CD3/CD28 stimulation (*Takada et al., 2011*). On the other hand, Pak2 plays essential roles in promoting functional maturation as well as KLF2 expression. These results indicate that Pak2-mediated functional maturation is required for the increased expression of KLF2. We propose that timed egress of thymocytes by KLF2 is tightly coordinated by a Pak2-dependent maturation process as a quality checkpoint mechanism. As a consequence, *Cd4*-Cre mediated Pak2 deletion results in severe T cell lymphopenia due to Pak2's multiple roles in T cell maturation and timed egress of thymocytes.

Since Pak2 is a serine/threonine kinase that phosphorylates multiple targets, it is of great interest to identify which substrates mediate Pak2's function in thymocyte development and maturation. Our results indicate that Pak2 facilitates actin cytoskeletal reorganization following TCR stimulation. Therefore, Pak2 may regulate thymic development by controlling TCR-mediated signaling strength by facilitating actin cytoskeletal reorganization. Among the known Pak2 substrates that are involved in cytoskeletal networks are: GEF-H1 (*Zenke et al., 2004*; *Kosoff et al., 2013*), ArhGAP15 (*Radu et al., 2013*), filamin A (*Vadlamudi et al., 2002*), Rho GDI (*DerMardirossian and Bokoch, 2006*), α and β PIX (*Koh et al., 2001*; *Shin et al., 2002*; *Rennefahrt et al., 2007*), regulatory myosin light chain (*Ramos et al., 1997*; *Chew et al., 1998*), myosin heavy chain (*Yamashita and May 1998*), tubulin cofactor B (*Vadlamudi et al., 2005*), LIMK (*Edwards et al., 1999*; *Dan et al., 2001*), and dynein light chain 1 (*Vadlamudi et al., 2004*). Moreover, it is likely that Pak2 is required for transcriptional responses that mediate the transition from the semi-mature to the mature stage since loss of Pak2 severely impaired this transition. It is not clear whether Pak2 is directly involved in regulating transcription responses by phosphorylating components of transcriptional machinery which mediate this transition, or whether Pak2 phosphorylates a signaling component that ultimately leads to the transcription response facilitating this transition. Further studies to determine the mechanism of Pak2's function in thymic development and maturation are warranted.

The Paks were the first Rho family GTPase-regulated kinases to be identified and are among the well-characterized effectors of Rac and Cdc42 (*Arias-Romero and Chernoff, 2008*). Here, we show that Pak2 is required for thymic development, including pre-TCR β selection and positive selection, and T cell activation, closely resembling the functions of Rac, Cdc42 or their activator Vav in T cell

development and activation. Together, our findings suggest that Pak2 is the primary effector for mediating the functions of Rac and Cdc42 in T cell development and activation.

## Materials and methods

### Mice

To generate a conditional knock-out of *Pak2* gene, a targeting vector was designed to flank exon 2 of *Pak2* gene with *loxP* sites as previously described (*Kosoff et al., 2013*). Mice with the *Pak2* floxed allele (*Pak2F/+*) were backcrossed at least nine times onto the C57BL/6 background. The Cd4-Cre (*Lee et al., 2001*) or Lck-Cre (*Hennet et al., 1995*) transgenic mice contain *Cd4* or *Lck* promoter sequences, respectively, driving the expression of a Cre recombinase gene. The *Cd4*-Cre or *Lck*-Cre transgenes were introduced *Pak2F/+* mice and backcrossed at least nine times onto the C57BL/6 backgrounds. *Pak2F/+;Cd4*-Cre or *Pak2F/+;Lck*-Cre mice in the C57BL/6 background were crossed to the OTII TCR transgenic mice (*Barnden et al., 1998*). C57BL/6, *Cd4*-Cre, *Lck*-Cre, OTII and BoyJ (B6 mice with the CD45.1+ congenic marker B6.SJL-Ptprca Pepcb/BoyJ) used for breeding were originally obtained from the Jackson Laboratory. All animals were housed in a specific pathogen-free facility at Northwestern University and University of California, San Francisco according to university and National Institutes of Health guidelines.

### PCR genotyping

The floxed *Pak2* mice were genotyped by genomic PCR isolated from tail clips. Amplification for the floxed *Pak2* gene was carried out by standard PCR protocol. Primers used to screen the floxed *Pak2* gene are; forward primer 5′-ATCTTCCCAGGCTCCTGACT, reverse primer 5′-TGAAGCTGCATCAATCTATTCTG. The Cre transgene in the *Cd4*-Cre or *Lck*-Cre mice was screened by standard PCR protocol using forward primer 5′-TGGGCGGCATGGTGCAAGTT and reverse primer 5′-CGGTGCTAACCA GCGTTTTC.

### Antibodies and reagents

Stimulatory armenian hamster anti-mouse CD3ε (2C11) antibody was purchased from BD Biosciences (San Jose, CA). Goat anti-Armenian hamster IgG (H and L chains) Abs and goat anti-rabbit IgG Abs conjugated to either PE or allophycocyanin were obtained from Jackson ImmunoResearch Laboratories (West Grove, PA). All antibodies for FACS analyses were obtained from BD Biosciences (San Jose, CA). Phospho-S6 Ribosomal protein (S235/236)(2F9), phospho-p70S6K(T389)(9234), and phospho-Erk1/2 (T202/Y204)(4377) antibodies were obtained from Cell Signaling Technologies (Danvers, MA). Phospho-PLCγ1 (Y783)(44-696G) antibody was purchased from Life Technologies (Grand Island, NY). Pak2 (TA306346) antibody is from Origene (Rockville, MD). Nur77(12.14) antibody was purchased from eBiosciences (San Diego, CA).

### Competitive repopulation experiments and bone marrow chimera

For mixed bone marrow (BM) chimera experiments, BM chimeras were generated by transferring a 1:1 mixture of BM cells from WT (CD45.1+CD45.2+) and *Pak2F/F;Cd4*-Cre mice (CD45.2+) with different congenic markers into lethally irradiated BoyJ recipient mice (B6.SJL-*Ptprca Pepcb*/BoyJ, CD45.1+). Mixed BM cells were resuspended in 1.0 ml ($10^7$ cells/ml) PBS and 200 µl ($2 \times 10^6$ mixed bone marrow cells) were injected i.v. into lethally irradiated BoyJ recipients. BoyJ recipients were irradiated with two doses of 600 rads, 3–5 hr apart. After 6–8 weeks, thymi, spleen, lymph nodes, and blood of reconstituted chimeras were harvested and used for flow cytometry analysis.

### Flow cytometry and data analysis

Single cell suspensions were prepared from thymi, lymph nodes (LN), and spleens. Fc receptors were blocked with rat anti-CD16/32 (2.4G2; BD Biosciences). 2,000,000 cells were stained with the indicated Abs and analyzed on a LSR II, FACSCanto or Fortessa (BD Biosciences) flow cytometry system. Forward and side scatter exclusion was used to identify live cells. Data analysis was performed using FlowJo (version 9.6.2) software (Tree Star). Statistical analysis and graphs were generated using Prism 6 (GraphPad Software).

### Fetal thymic organ culture

Fetal thymi from embryos at embryonic day 16 were explanted and cultured on the top surface of a trans-well. The bottom surface of the trans-well was submerged in 10% FBS containing media in the

bottom well, but the thymi were exposed to 5% CO2 during culture. The media in the bottom well was changed every day up to 7 days. In the end of 7 days of culture, single cell suspensions from the fetal thymi were prepared and used for flow cytometry.

## CFSE (5-(and −6)-carboxyfluorescein diacetate succinimidyl ester) dilution assay

Plates (24 well) were coated with anti-CD3 (1 µg/ml) or anti-CD3 (1 µg/ml) and anti-CD28 (10 µg/ml) for overnight at 4°C. Single cell suspensions of thymocytes were washed with PBS without $Ca^{2+}$ and $Mg^{2+}$ twice, then resuspended at $2 \times 10^7$ cells/ml. A CFSE solution was added to the cells to a final concentration of 2.5 µM for 8 min at room temperature. After 8 min, fetal calf serum was directly added to the CFSE labeled cells and incubated for 1 min. Cells were washed twice with 5 min interval. 2,000,000 of CFSE labeled cells were added to pre-coated plates. After 72 hr, cells were harvested and stained for surface markers as well as DAPI (4',6-diamidino-2-phenylindole). Data were collected on the BD LSRII or FACSCanto flow cytometer and analyzed using FlowJo software.

## Intracellular staining for phospho-S6

Thymocytes were harvested, washed in FACS buffer and stained for surface markers on ice. Following surface staining, cells were washed and fixed with Fixation/Permeabilization buffer (eBiosciecne) for 14 hr at 4°C following manufacturer's instruction. Cells were washed with 1X Permeabilization buffer (eBioscience) and stained with FITC-anti-phospho-S6 (S235/236) (Cell signaling technologies) for 1 hr. Cells were washed twice in 1X permeabilization buffer (eBioscience) and resuspended in FACS buffer. Data were collected on the BD LSRII or FACSCanto flow cytometer and analyzed using FlowJo software.

## Intracellular staining for Nur77

2,000,000 thymocytes were added to each well of 24-well plates that were coated with anti-CD3ε. After stimulation, cells were harvested using ice-cold PBS, transferred to FACS tubes, fixed for 10 min with equal volume of Cytofix (BD Biosciences) to terminate stimulation, and permeabilized using 95% ice-cold methanol (EMS, Hatfield, PA) for 30 min. Cells were washed twice, re-hydrated in staining buffer (2% BSA in PBS) for 30 min, then stained with Nur77-PE (eBioscience) Ab in staining buffer for 1 hr. After Nur77 staining, cells were washed twice, stained with surface markers for 30 min. Data were collected on FACSCanto or LSRII flow cytometer (BD Biosciences) and analyzed using FlowJo software.

## Thymocyte sorting and quantitative PCR

Single cell suspensions from thymi were prepared and stained using CD4, CD8, CD69, CD62L, CD24 and Qa2 surface markers as well as DAPI (4',6-diamidino-2-phenylindole) to identify DP, semi-mature (CD69$^{hi}$CD62L$^{low}$ or CD24$^{hi}$Qa2$^{low}$) and mature (CD69$^{low}$CD62L$^{hi}$ or CD24$^{low}$Qa2$^{hi}$) CD4SP thymocytes. Since the CD69$^{low}$CD62L$^{hi}$ mature CD4SP thymocyte numbers were significantly reduced from *Pak2$^{F/F}$;Cd4*-Cre mice, we sorted CD69$^{low}$CD62L$^{med}$ CD4SP thymocytes from *Pak2$^{F/F}$;Cd4*-Cre mice. Alternatively, we used CD24 and Qa2 surface expression to sort the semi-mature CD24$^{hi}$Qa2$^{low}$ CD4SP thymocytes or the CD24$^{low}$Qa2$^{hi}$ CD4SP thymocytes from *Pak2$^{F/F}$* or *Pak2$^{F/F}$;Cd4*-Cre mice in some experiments. Thymocytes were sorted using a MoFlo cell sorter. Cells were lysed and total RNA was prepared using RNeasy and QIAshredder kit (Qiagen, Valencia, CA). First strand cDNA were synthesized using SuperScript III First-Strand Synthesis System (Life Technologies). cDNA was analyzed in triplicate by QPCR amplification by incorporation of Platinum SYBR Green or Fast SYBR Green with a StepOnePlus Real-Time PCR System (Applied Biosystems, Life technologies), and results are presented relative to the expression of *Hprt* (encoding murine hypoxanthine phosphoribosyltransferase). Data were analyzed by comparative quantification. PCR primer pairs are as follows: HPRT forward, 5'-AGTCCCAGCGTCGTGATTAGC-3'; HPRT reverse, 5'-GAGCAAGTCTTTCAGTCCTGTCC-3'; KLF2 forward 5'-CTAAAGGCGCATCTGCGTA-3'; KLF2 reverse 5'-TAGTGGCGGGTAAGCTCGT-3'; S1P1 forward 5'-GTGTAGACCCAGAGTCCTGCG-3'; S1P1 reverse, 5'-AGCTTTTCCTTGGCTGGAGAG-3'.

## Actin polymerization assay

Amounts of polymerized actin in thymocytes were measured as previously described (*Phee et al., 2010*).

## Rac1 activation assay

Specific isolation of Rac1-GTP was performed as described previously (*Phee et al., 2010*).

## T cell spreading assay and confocal microscopy

Untouched CD4SP thymocytes were enriched by removing DP and CD8 SP thymocytes using CD8 MACS beads (Miltenyi Biotec, San Diego, CA) and plated onto cover slips that are coated with anti-CD3 (10 µg/ml) (*Bunnell et al., 2001*). Following 60 min of stimulation, cells were fixed, permeabilized, stained with Alexa Flour488 Phalloidin, and mounted using ProLong Gold Antifade reagent (Life Technologies) as previous described (*Phee et al., 2010*). Confocal images were collected using an A1R Resonant Scanning Multispectral Confocal Microscope (Nikon Instruments Inc, Melville, NY). 40X (N.A. 1.3) Plan Fluor lens or 100X (N.A. 1.45) PlanApo lens were used. Z-stack images were collected from the contact site where T cells touch the cover slides (bottom) to the top of the cells where no signals were detected (top). In most cases, total distance of z stacks was 11–12 µm and Nyquist-compatible 0.25 µm step size was used to collect z-stack images. The contact areas where the cell spreads were measured by using the NIS-Elements Viewer.

## Acknowledgements

The authors would like to thank the Phee and Weiss laboratories for discussions. All microscopy studies were performed at Northwestern University Cell Imaging Facility. This work was supported grants from US National Institutes of Health 5K01AR059754 (HP) and R01 CA142928 (JC), as well as the Howard Hughes Medical Institute (AW).

## Additional information

### Funding

| Funder | Grant reference number | Author |
|---|---|---|
| National Institutes of Health | K01 AR059754 | Hyewon Phee |
| National Institutes of Health | R01 CA142928 | Jonathan Chernoff |
| Howard Hughes Medical Institute | | Arthur Weiss |

The funders had no role in study design, data collection and interpretation, or the decision to submit the work for publication.

### Author contributions

HP, Conception and design, Acquisition of data, Analysis and interpretation of data, Drafting or revising the article; BBA-Y, OP, KLO'H, SGF, MM, DC, Acquisition of data, Analysis and interpretation of data, Drafting or revising the article; MR, RK, JC, Analysis and interpretation of data, Drafting or revising the article, Contributed unpublished essential data or reagents; AW, Conception and design, Analysis and interpretation of data, Drafting or revising the article

### Ethics

Animal experimentation: This study was performed in strict accordance with the recommendations in the Guide for the Care and Use of Laboratory Animals of the National Institutes of Health. All of the animals were handled according to approved institutional animal care and use committee (IACUC) protocol (#2012-2851) of Northwestern University.

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
