## [Decision Letter]

Thank you for sending your work entitled “Pak2 is required for actin cytoskeleton remodeling, TCR signaling, and normal thymocyte development and maturation” for consideration at *eLife*. Your article has been favorably evaluated by a Senior editor, a Reviewing editor, and 2 reviewers.

The Reviewing editor and the reviewers discussed their comments before we reached this decision, and the Reviewing editor has assembled the following comments to help you prepare a revised submission.

This manuscript explains the signal transduction pathways that control the development of T cells in the thymus using selective deletions of the Set/Thr kinanse Pak2. The data show that Pak2 is important for TCR-beta selection and thymocyte maturation in vivo. Overall the experiments are well performed and the conclusions are justified by the data. Both reviewers felt that some discussion of the options for Pak2 substrates in the context of T cell development is necessary to place this study in the context of other information concerning Pak 1 and Pak2 in the literature.

Minor comments:

1) The data in Figure 1 and Figure 1—figure supplement 1 are used to show that the deletion of Pak in thymocytes causes a peripheral lymphopenia. Cell numbers should be shown in the main Figure rather than as supplemental data.

2) In the text (in the context of Figure 3), authors mention as data not shown that the levels of expression of CD69 in DP and CD4SP are normal in Pak2f/f;CD4-Cre, but data in Figure 4 show increased CD69 expression in Pak2f/f;CD4-Cre CD4SP. Which is correct?

3) The authors state that deletion of Pak2 at the DN2 stage blocks positive selection. Given their later results this statement should be modified to say that the decreased numbers of SPs in the *Lck*-Cre pakfl/flOTII mice could be due to decreased selection or decreased maturation.

4) Levels of TCRb or CD3 expression in Pak2^f/f^;CD4-Cre CD4SP. Authors include the cell surface expression of TCRb in CD4-SP in the mixed bone marrow chimera experiments (Figure 6), but not for the Pak2^f/f^;CD4-Cre CD4SP mice (Figure 4). Inclusion of that data will be valuable to prove that the defects on TCR signalling shown in Figure 7 are not due to impaired levels of TCR expression at cell surface.

5) In Figure 7 the authors claim that the loss of Pak1 prevents expression of KLF2 and S1P1. However, these molecules are expressed at the highest level in the most mature SPs subsets. Hence the fact that there is a decrease in KLF2 and S1P1 in Pak2 deficient CD4s probably reflects that this mature subset is not present. This point should be discussed.

6) The authors use the phosphorylation of the ribosomal S6 subunit as a readout of the mTOR signalling pathway in CD4SP. However, Salmond et al show that pS6 is controlled by MAPK-Rsk and mTOR. Thus, the phosphorylation of S6 can measure either pathway and more work is required to verify which one (e.g., rapamycin sensitivtity of S6 phosphorylation indicates the mTOR pathway).

7) The ordering of panels in Figure 7 is a bit confusing as it does not match the order in the text

8) There should be some discussion about possible Pak2 substrates responsible for these defects.

---

## [Author Response]

*Both reviewers felt that some discussion of the options for Pak2 substrates in the context of T cell development is necessary to place this study in the context of other information concerning Pak 1 and Pak2 in the literature*.

We have added a paragraph in the Discussion to specifically address the options for Pak2 substrates that could influence T cell development.

*1) The data in*
Figure 1
*and*
Figure 1—figure supplement 1
*are used to show that the deletion of Pak in thymocytes causes a peripheral lymphopenia. Cell numbers should be shown in the main Figure rather than as supplemental data*.

We changed Figure 1 to show the cell numbers. We added panels Figure 1, which show the cell numbers of CD4 and CD8 T cells and CD62L^hi^CD44^low^(naïve) CD4 and CD8 T cells. To save space, we removed FACS profiles of *Pak2*^*F/+*^;*Lck*-Cre heterozygous mice in Figure 1. The numbers of CD3, CD19 and CD62L^low^CD44^hi^(memory) CD4 and CD8 T cells are provided in Figure 1—figure supplement 1.

*2) In the text (in the context of*
Figure 3*), authors mention as data not shown that the levels of expression of CD69 in DP and CD4SP are normal in Pak2f/f;CD4-Cre, but data in*
Figure 4
*show increased CD69 expression*
*in Pak2f/f;CD4-Cre CD4SP. Which is correct?*

We have corrected the text to read “Expression of CD5, CD69, and CD3 on DP thymocytes were similar comparing *Pak2^F/F^* and *Pak2^F/F^;Cd4-*Cre mice”.

*3) The authors state that deletion of Pak2 at the DN2 stage blocks positive selection. Given their later results this statement should be modified to say that the decreased numbers of SPs in the* Lck-*Cre pakfl/flOTII mice could be due to decreased selection or decreased maturation*.

This is a valid point. We now provide data supporting the point that maturation of CD4SP thymocytes from *Pak2^F/F^*;*Lck*-Cre or OTII+;*Pak2^F/F^*;*Lck*-Cre are also inhibited in Figure 4—figure supplement 1. We added these data and modified the text, stating that the decreased numbers of CD4SP thymocytes from *Pak2^F/F^*;*Lck*-Cre or OTII+;*Pak2^F/F^*;*Lck*-Cre could be due to decreased positive selection or impaired maturation.

*4) Levels of TCRb or CD3 expression in Pak2*^*f/f*^*;CD4-Cre CD4SP. Authors include the cell surface expression of TCRb in CD4-SP in the mixed bone marrow chimera experiments (*Figure 6*), but not for the Pak2*^*f/f*^*;CD4-Cre CD4SP mice (*Figure 4*). Inclusion of that data will be valuable to prove that the defects on TCR signalling shown in*
Figure 7
*are not due to impaired levels of TCR expression at cell surface*.

We added figures showing that expression levels of CD3 and TCRβ are similar between *Pak2^F/F^* and *Pak2^F/F^*;*Cd4*-Cre mice in Figure 4.

*5) In*
Figure 7
*the authors claim that the loss of Pak1 prevents expression of KLF2 and S1P1. However, these molecules are expressed at the highest level in the most mature SPs subsets. Hence the fact that there is a decrease in KLF2 and S1P1 in Pak2 deficient CD4s probably reflects that this mature subset is not present. This point should be discussed*.

This is a valid point. Since Pak2 is required for the transition from the semi-mature to the mature stage, the decrease in KLF2 and S1P1 mRNA expression may reflect the lack of the mature CD4SP thymocyte subset in the absence of Pak2. However, we also observed a reproducible decrease of KLF2 mRNA expression even at the semi-mature stage of CD4SP thymocytes from *Pak2^F/F^;Cd4-*Cre mice, suggesting Pak2 contributes to expression of KLF2 even at an earlier stage than the mature stage, when CD4SP thymocytes are just starting to upregulate KLF2. Therefore, we have now added a new panel (right) in Figure 7, showing the relative expression of KLF2 in the absence of Pak2 at the semi-mature stage. S1P1 is minimally expressed at the semi-mature stage compared to the mature stage. Although expression levels of S1P1 were minimal, we consistently saw a decrease in S1P1 mRNA expression in the semi-mature CD4SP thymocytes of *Pak2^F/F^;Cd4-*Cre mice compared to *Pak2^F/F^* mice. We also have added a new panel (right) in Figure 7, showing relative expression of S1P1 in the absence of Pak2 at the semi-mature stage. Therefore, we believe that, although the decrease in mRNA expression of KLF2 or S1P1 may reflect the lack of the mature subset, Pak2 regulates expression of KLF2 even at the earlier semi-mature stage when KLF2 expression is just beginning to increase.

We have now added text explaining this in detail: “Since Pak2 is required for the transition from the semi-mature to the mature stage, the decrease in KLF2 and S1P1 mRNA expression may reflect the lack of the mature CD4SP thymocytes in the absence of Pak2. However, we observed a reproducible decrease in KLF2 expression in the semi-mature CD4SP thymocytes of *Pak2^F/F^;Cd4-*Cre mice (Figure 7, right panel), suggesting Pak2 contributes to expression of KLF2 when CD4SP thymocytes are just beginning to upregulate KLF2.”

*6) The authors use the phosphorylation of the ribosomal S6 subunit as a readout of the mTOR signalling pathway in CD4SP. However, Salmond et al show that pS6 is controlled by MAPK-Rsk and mTOR. Thus, the phosphorylation of S6 can measure either pathway and more work is required to verify which one (e.g., rapamycin sensitivtity of S6 phosphorylation indicates the mTOR pathway)*.

This is a good suggestion. To assess the role of the mTOR and MAPK-RSK pathways in TCR-induced p70S6K and S6 phosphorylation, we performed inhibitor studies using an mTOR inhibitor, rapamycin, and a MEK inhibitor, UO126.

First, we confirmed that efficacy of the MEK inhibitor UO126 (10 μM). WT thymocytes were pretreated with UO126 (10 μM) for 30 min, then stimulated for 30 and 60 minutes with plate-bound anti-CD3 antibodies. Cells were lysed following stimulation, and cell lysates were subjected to immunoblotting analysis using anti-phosphospecific antibodies. Phosphorylation of Erk1/2 and RSK was completely abrogated as Erk and RSK have been shown to be downstream targets of MEK (Figure 7—figure supplement 2). We also treated cells with rapamycin to determine whether inhibition of mTOR affects phosphorylation of Erk1/2 or RSK. We found that rapamycin did not substantially affect phosphorylation of Erk1/2 or RSK (Figure 7—figure supplement 2).

Next, we tested that whether the mTOR inhibitor rapamycin reduces TCR-induced phosphorylation of p70S6K at the T389 site as this site has been shown to be dependent on mTOR activity as shown in Salmond et al. Phosphorylation of p70S6K at the T389 site after 30 and 60 minutes of anti-CD3 stimulation was completely inhibited by rapamycin treatment, suggesting phosphorylation of p70S6K at T389 site is dependent on mTOR activity (Figure 7—figure supplement 2).

To determine whether phosphorylation of p70S6K at T389 site is dependent on the MAPK pathway, we measured phosphorylation of p70S6K at T389 site following MEK inhibitor UO126 treatment (Figure 7—figure supplement 2). We saw minimal reduction on phosphorylation of p70S6K at T389 in cells treated with the MEK inhibitor UO126, while rapamycin completely inhibited phosphorylation of p70S6K at T389, suggesting phosphorylation of p70S6K at T389 is mediated by mTOR-dependent, not by a MAPK-dependent pathway.

We showed that Pak2 is required for optimal phosphorylation of S6 at S235/S236 (Figure 7), and that phosphorylation of these sites is highly dependent on mTOR (Figure 7—figure supplement 2). To determine whether reduced phosphorylation of S6 at S235/S236 in the absence of Pak2 is due to inhibition of mTORC1-mediated pathway, we examined phosphorylation of p70S6K at the T389 site because phosphorylation of this site was completely dependent on mTORC1-mediated pathway but independent of MAPK-mediated pathway (Figure 7—figure supplement 2). Phosphorylation of p70S6K at T389 was also decreased following 30 min and 1 hour of plate-bound CD3 stimulation in the absence of Pak2 (Figure 7). Therefore, we conclude that Pak2 participates in mTORC1-mediated activation of p70S6K and phosphorylation of S6. We added a section describing this finding in the manuscript.

*7) The ordering of panels in*
Figure 7
*is a bit confusing as it*
*does not match the order in the text*

We corrected the order of panels in Figure 7 so that they match the order in the text.

*8) There should be some discussion about possible Pak2 substrates responsible for these defects*.

We added text discussing this issue in the following paragraph: “Since Pak2 is a serine/threonine kinase that phosphorylates multiple targets, it is of great interest to identify which substrates mediate Pak2’s function in thymocyte development and maturation … Further studies to determine the mechanism of Pak2’s function in thymic development and maturation are warranted.”